# Improving gene function predictions using independent transcriptional components

Carlos G. Urzúa-Traslaviña [1,3], Vincent C. Leeuwenburgh [1,2,3], Arkajyoti Bhattacharya [1],
Stefan Loipfinger [1], Marcel A. T. M. van Vugt [1], Elisabeth G. E. de Vries [1] & Rudolf S. N. Fehrmann [1✉]

The interpretation of high throughput sequencing data is limited by our incomplete functional understanding of coding and non-coding transcripts. Reliably predicting the function of such transcripts can overcome this limitation. Here we report the use of a consensus independent component analysis and guilt-by-association approach to predict over 23,000 functional groups comprised of over 55,000 coding and non-coding transcripts using publicly available transcriptomic profiles. We show that, compared to using Principal Component Analysis, Independent Component Analysis-derived transcriptional components enable more confident functionality predictions, improve predictions when new members are added to the gene sets, and are less affected by gene multi-functionality. Predictions generated using human or mouse transcriptomic data are made available for exploration in a publicly available web portal.

[1] Department of Medical Oncology, University Medical Center Groningen, University of Groningen, Groningen, The Netherlands. [2] The Stratingh Institute for Chemistry, University of Groningen, Groningen, The Netherlands. [3]These authors contributed equally: Carlos G. Urzúa-Traslaviña, Vincent C. Leeuwenburgh. ✉email: r.s.n.fehrmann@umcg.nl

Modern high-throughput experiments enable researchers to determine the association between each individual gene and a phenotype. Currently, gene set enrichment analysis (GSEA) is used to interpret which biological processes are associated with the phenotype of interest[1]. Gene sets are collections of genes that share a common characteristic (e.g., biological function). Overrepresentation of gene sets in the top associated genes can be used to identify which biological processes are driving the phenotype. Although gene sets are continuously updated, they are still far from complete. This can bias and hamper the interpretation of high-throughput experiments[2].

A strategy to reduce this limitation is to extend gene sets by providing a gene set membership likelihood for every gene. This can be accomplished by comparing gene co-expression patterns within a guilt-by-association (GBA) strategy. For example, if there are 50 genes that are known to be involved in DNA repair, a gene that is strongly co-expressed with these 50 is likely to be involved in DNA repair as well. A potential limitation arises when a gene is involved in multiple biological processes[3]. Each biological process is the result of the coordinated expression of multiple genes, which together represent a transcriptional footprint. One of these biological processes might have a transcriptional footprint that dominates the observed gene expression profile and therefore overshadows the other biological processes with more subtle transcriptional footprints[4]. Therefore, relying on co-expression may bias GBA predictions for genes involved in multiple biological processes towards those biological processes with dominant transcriptional footprints.

A strategy to overcome this limitation is to first apply principal component analysis (PCA) to the gene expression input data. Applying PCA results in a smaller set of new variables called principal components (PCs). These PCs capture most of the variance of the original input data. In the context of gene expression data, PCs are referred to as transcriptional components (TCs). Both dominant and more subtle transcriptional footprints of biological processes can be captured by PCA-TCs[4]. Using PCA-TCs in a GBA strategy improves gene set membership predictions compared to regular gene to gene co-expression patterns[5]. However, two or more PCA-TCs may still possess a mutual nonlinear correlation. This correlation reflects the transcriptional footprint of a biological process that is partially captured in multiple PCA-TCs. Segregating nonlinear correlations into individual TCs may further improve gene set membership predictions. Independent component analysis (ICA) can enable the estimation of ICA-TCs that capture strong and subtle transcriptional footprints while minimizing their mutual nonlinear correlation[6].

In this study, we utilize three large messenger RNA (mRNA) expression datasets originating from microarray and RNA-sequencing (RNA-seq) platforms, and 16 gene set collections in a GBA strategy using ICA-TCs to infer gene set memberships for up to 58,433 individual genes. We show that using ICA-TCs (1) provides more confident functionality predictions, (2) improves predictions when new members are added to the gene sets, and (3) is less affected by gene multifunctionality in comparison with PCA-TCs.

## Results

**Data acquisition**. To generate our human microarray input dataset, all publicly available unprocessed microarray expression data generated with the Affymetrix HG-U133 Plus 2.0 platform were downloaded from the Gene Expression Omnibus (GEO)[7]. After preprocessing and quality control, a total of 106,462 samples were included for further analysis. The R package "jetset" (version 3.4.0) served to obtain one-to-one mapping between genes and the most representative probe sets on the Affymetrix HG-U133 Plus 2.0 platform and resulted in unique expression measurements for 19,635 genes (R jetset package v3.4.0)[8]. Gene set collections ($n = 16$) were retrieved from The Human Phenotype Ontology (The Monarch Initiative), the Mammalian Phenotypes (Mouse Genome Database), and 14 collections from the Molecular Signatures Database v6.2[9–11]. These gene set collections contain 23,372 gene sets in total.

**Consensus ICA captures the variance of the input dataset**. Consensus ICA (c-ICA) was performed on the input dataset to calculate ICA-TCs (Fig. 1). In ICA, a preprocessing technique called whitening is applied to make the estimation more time efficient by reducing the dimensionality of the input dataset. Whitening was achieved with PCA on the covariance matrix between the genes and resulted in 817 PCs that captured 90% of the total variance in gene expression observed in the human microarray input dataset. Subsequently, using these 817 PCs as input for c-ICA, we obtained 523 ICA-TCs with a credibility index $>0.5$.

**Framework for generating predictions using ICA-TCs**. The c-ICA method provides a mixing matrix (MM) together with the ICA-TCs. In the MM each column corresponds to an ICA-TC and each row corresponds to a gene. Each weight in the MM describes the effect of a latent transcriptional regulating factor on the expression level of a gene. The vector of weights of an individual gene in the MM is referred to as the transcriptional regulatory "barcode." For each of the 23,372 gene sets, an average transcriptional regulatory barcode was calculated by taking the mean vector of weights in the MM of genes that are members of the gene set. Next, the distance correlations between the transcriptional regulatory barcode of all combinations of genes and gene sets were calculated[12]. A strong positive correlation indicates that the transcriptional regulation of an individual gene is similar to the average transcriptional regulation of members of the gene set. Gene sets that capture biological processes with highly co-regulated genes generate consistent regulatory barcodes. Therefore, genes with strong correlation have a high likelihood of belonging in those gene sets even if they are not currently members. For example, BRCA2 is a DNA repair protein with a strong co-regulation with the cell cycle KEGG (Kyoto Encyclopedia of Genes and Genomes) gene set (prediction score = 15.53). Even though BRCA2 does not directly regulate the cell cycle, progression through the G2–M checkpoint depends on the successful repair of all DNA damaged during replication. Therefore, when viewed through co-regulation BRCA2 participates in two biological processes, namely, DNA repair and cell cycle progression.

A permutation strategy in combination with a Gaussian kernel density estimator enabled the construction of a null distribution for each gene-to-gene set combination. The permutation tests determined the significance of the distance correlation without assuming an underlying shape for the null distribution of each method. Resulting $p$ values were transformed to a Z-score, hereafter referred to as prediction score (see Supplementary Notes). The prediction scores obtained using the human microarray dataset are made available on a portal at http://genetica-network.com. This interface allows users to rank the scores of all genes that predict membership to a particular gene set (gene set perspective) or the scores of all gene sets that predict membership of a particular gene (gene perspective).

**Using ICA-TCs results in stronger predictions than PCA-TCs**. Data-driven Expression Prioritized Integration for Complex Traits (DEPICT) and the more recent GeneNetwork Assisted

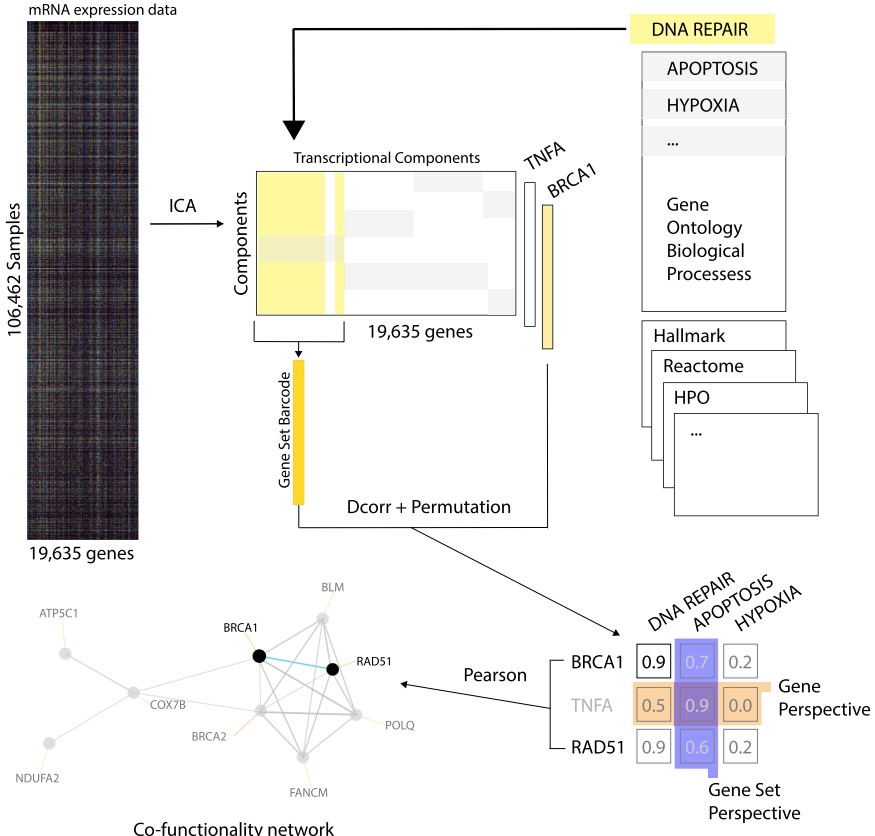

**Fig. 1 Overview of the guilt-by-association methodology to calculate prediction scores.** Transcriptional components (TCs) are calculated from a large mRNA expression dataset using c-ICA or PCA. For every gene set in a gene set collection, a vector of means (transcriptional regulatory barcode) is calculated by taking the mean vector of weights in the mixing matrix of genes that are members of the gene set. The correlation of each gene and each barcode is calculated using distance correlation and a Z-transformed *p* value is estimated by performing a permutation test. The resulting Z-transformed *p* values constitute prediction scores that can be interpreted as a ranking of gene set memberships predicted for a gene (gene perspective). Alternatively, they can be interpreted as a ranking of genes predicted as members of a single gene set (gene set perspective). Finally, a gene-to-gene correlation matrix is calculated from the prediction scores and used to cluster genes in the force-directed layout of the co-functionality network visualization.

**Table 1 Guilt-by-association methods.**

|  | DEPICT and GADO | GENETICA (ours) |
|---|---|---|
| Strategy | Guilt-by-association prediction of gene functions using gene sets | Guilt-by-association prediction of gene functions using gene sets |
| Transcriptional components | Principal component analysis | Independent component analysis |
| Barcode | Comparison of eigenvector scalars of member genes and nonmember genes using Welch *T* test | Average of all mixing matrix weights of member genes |
| Distance metric | Pearson correlation | Distance correlation |
| Permutation test | No, parametric *p* values used | Yes, permutation *p* values used |
| Reference | 5,13 | This study |

Diagnostic Optimization (GADO) are two successful methods that utilize gene functional predictions obtained with a PCA-TC-based GBA method[5,13]. DEPICT and GADO use predictions for biological interpretation of genome-wide association studies and to improve the discovery of disease-causing genes from exome sequencing, respectively (Table 1). A comparison between the predictions obtained with a GBA method using ICA-TCs and PCA-TCs was performed. To ensure a fair comparison, the whitened matrix that was used to run c-ICA served as input for the PCA-TC-based method. Using the whitened matrix as input ensures that both methods have the same amount of variance in the expression data available for generating TCs.

First, genes that were already assigned to gene sets were analyzed. A gene is expected to get a high prediction score for a gene set of which it is currently a member. For each gene set, a median prediction score for the subset of current member genes was calculated as well as for the subset of genes that are not assigned to any gene set in that collection (nonmember genes) (Supplementary Data 1). Median prediction scores of member genes were higher compared with nonmember genes in both the PCA-TC- and the ICA-TC-based method. However, ICA-TC-based predictions outperformed PCA-TC by obtaining higher median prediction scores for current member genes in all gene set collections (area under the curves (AUCs) calculated from

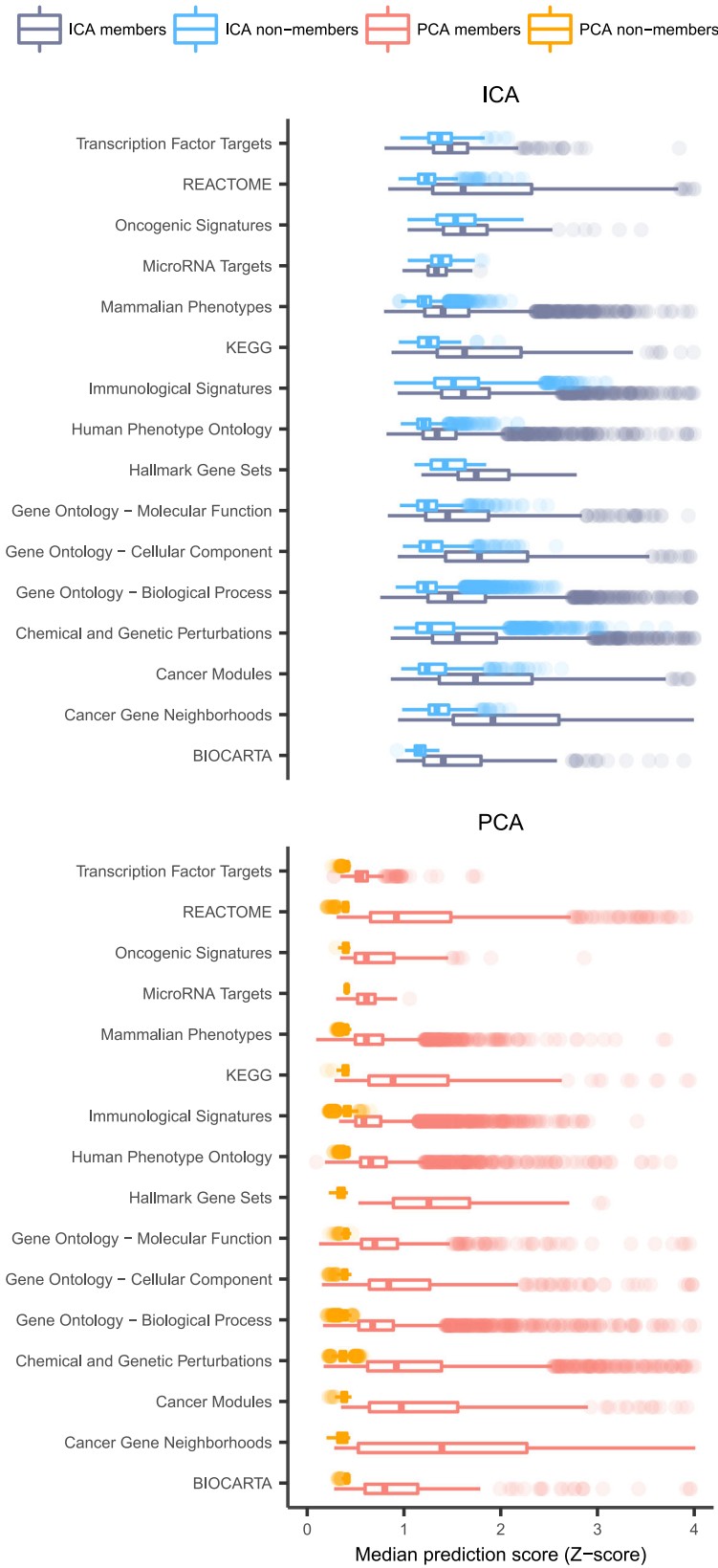

**Fig. 2 Distribution of prediction scores for both the ICA-TC- and PCA-TC-based method.** Boxplot of median prediction scores (x-axis) calculated from the ICA-TC-based (blue) and PCA-TC-based (red) methodology for each of the 16 gene set collections (y-axis). Median prediction scores are calculated separately for each gene set using both the ICA-TC- and the PCA-TC-based method for member (saturated) and nonmember genes (less saturated) and plotted side to side. Prediction scores of ICA-TC-based method are higher for both the members and the nonmember gene subset. ICA-TC-based predictions outperformed PCA-TC by obtaining higher median prediction scores for current member genes in all gene set collections (area under the curves (AUCs) calculated from two-sided Mann–Whitney U test range 0.7–0.99). Hinges of boxes represent second and third quartiles and whiskers extend by half that interquartile range. Center of box corresponds to median.

two-sided Mann–Whitney $U$ test range 0.7–0.99; Fig. 2). Moreover, median prediction scores for nonmember genes were also higher for ICA-TC- than PCA-TC-based predictions (AUCs all >0.99). These results indicate that the ICA-TC-based method provides predictions with higher confidence than the PCA-TC-based method.

To explore if a subset of gene sets is better predicted with the PCA-TC-based method the differences between gene set prediction scores of both PCA and ICA methods were calculated (delta). Gene sets with deltas higher than zero were defined as ICA-improved and below zero as PCA-improved gene sets. We observed that 811/23,413 (3.4%) gene sets belonged to the PCA-improved subset (Supplementary Fig. 1). Gene sets belonging to the PCA-improved subset had fewer members (median = 23) than the ICA-improved subset (median = 57).

**ICA-TCs outperform PCA-TCs in predicting future ontologies**. The predictive power of the PCA-TC- and ICA-TC-based GBA prediction strategies was further evaluated by generating predictions using a previous version of three Gene Ontology (GO) gene set collections (Molecular Signatures Database C5 v3.0). Genes that are not members in v3.0 are expected to obtain high prediction scores if they got added as members to those gene sets in later version updates. In addition, such a gene would be expected to have an equal or higher prediction score when predictions are generated using the newer gene set versions with updated membership information. This would indicate the capacity of the method to predict future gene set memberships and to improve predictions when updated gene set membership information is available.

Genes that were added as member of a gene set after v3.0 are referred to as updated member genes. Median prediction scores for these updated member genes were calculated with the ICA-TC- and PCA-TC-based method (Fig. 3). This was done independently with v3.0 and v6.2 as input for the transcriptional barcode generation step. Only gene sets that got new genes added in between v3.0 and v6.2 were included in this analysis. Prediction scores for the updated member genes with v3.0 barcodes were higher for ICA-TC- compared to the PCA-TC-based method (AUC ranging from 0.90 to 0.96). This indicates that the ICA-TC-based method outperforms the PCA-TC method at predicting future gene memberships in the GO gene set collection.

Surprisingly, for the PCA-TC-based method, scores of the updated member genes were lower when using v6.2 barcodes than when using v3.0 barcodes as input (AUC ranging from 0.19 to 0.34). In contrast, using v6.2 as input did improve prediction scores when applying ICA-TC-based method (AUCs ranging from 0.65 to 0.71). This indicates that the ICA-TC- but not the PCA-TC-based method is able to improve predictions when updated membership information is available for the GO gene set collections.

**Gene membership frequencies do not drive prediction scores**. Previously, it was shown that multifunctionality, rather than association, can be a main driver of functional predictions when using GBA methodology[14]. Here, the degree of multifunctionality of a gene is defined as a function of the number of gene sets of which it is currently a member[14]. The degree of multifunctionality of a gene might have a strong correlation with the gene set prediction scores of that gene. This would indicate that the likelihood of a gene being a member of the gene set, as expressed by the prediction score, can largely be explained by the

degree of multifunctionality and not by the predictions obtained with the GBA strategy.

Per gene set collection, the degree of multifunctionality was calculated for all genes (see "Methods"). The associations between the degree of multifunctionality and the prediction scores (Fig. 4 and Supplementary Data 2) were lower for the ICA-TC-based method than the PCA-TC-based method (AUCs range 0.03–0.47). This means that ICA-TC-based predictions are less driven by the membership frequencies of a gene and therefore are relying more on the GBA strategy. The predictions of some gene set collections showed on average higher association with multifunctionality. The most affected gene set collections were either constructed based on predicted targets of transcriptional regulators (microRNA and transcription factor targets) or obtained from signatures of published perturbation experiments (Chemical and Genetic Perturbations and Immune signatures). In conclusion, the association between the degree of multifunctionality and prediction scores is very limited for most gene sets for both the ICA-TC and PCA-TC method.

**Mouse and human prediction scores are correlated**. In addition, ICA-TC-based gene function predictions were generated for mouse microarray samples (GPL1261 platform; $n = 25,585$; genes = 18,425) for 16 gene set collections ($n = 23,128$). We correlated the prediction scores between the unique ortholog mapping genes ($n = 14,589$) between human and mouse microarray datasets for every matching gene set collection to investigate the concordance between prediction scores (Supplementary Fig. 2). A subset of 10,974/14,589 (75.2%) mouse genes have a correlation >0.3 to their human orthologs based on their prediction scores for at least one of the gene set collections. GO Cellular Component and Cancer Modules gene set collections showed the highest median Spearman correlations (0.155 and 0.154, respectively). The low correlating gene predictions across species could be explained by the smaller number of samples ($n = 25, 585$) and the more limited sample heterogeneity available in the mouse dataset compared to the human microarray dataset ($n = 106,462$). These results show that many genes may be similarly regulated in both species. The mouse prediction scores have been made available at http://www.genetica-network.com.

**Current gene sets do not capture all co-regulation clusters**. When using a GBA strategy with mRNA profiles, high prediction scores for a biological process depend on three factors. First, the existence of at least one gene set that covers that process. Second, if such a gene set contains all relevant genes involved in that biological process. Third, the degree of co-regulation among the genes of that biological process as reflected by possessing similar transcriptional regulatory barcodes. The existence of co-regulated biological processes that are underrepresented in current gene set collections was evaluated with the human microarray-based prediction scores. Hierarchical clustering was applied to the transcriptional regulatory barcodes of individual genes (see "Methods" for details), which resulted in 173 clusters of co-regulated genes (Supplementary Data 3). The size of the clusters ranged between 10 and 389, which is similar to the size range 23,372 gene sets used in our ICA-TC-based GBA method.

For each cluster, the similarity among transcriptional regulatory barcodes was defined as the median pairwise correlation between gene members. This is referred to as the cluster transcriptional similarity score. In addition, for each member gene, the maximum prediction score obtained in any of the 23,372 gene sets in our framework was obtained. Then, a median per cluster was calculated, referred to as cluster predictability score. A correlation ($R = 0.59$)

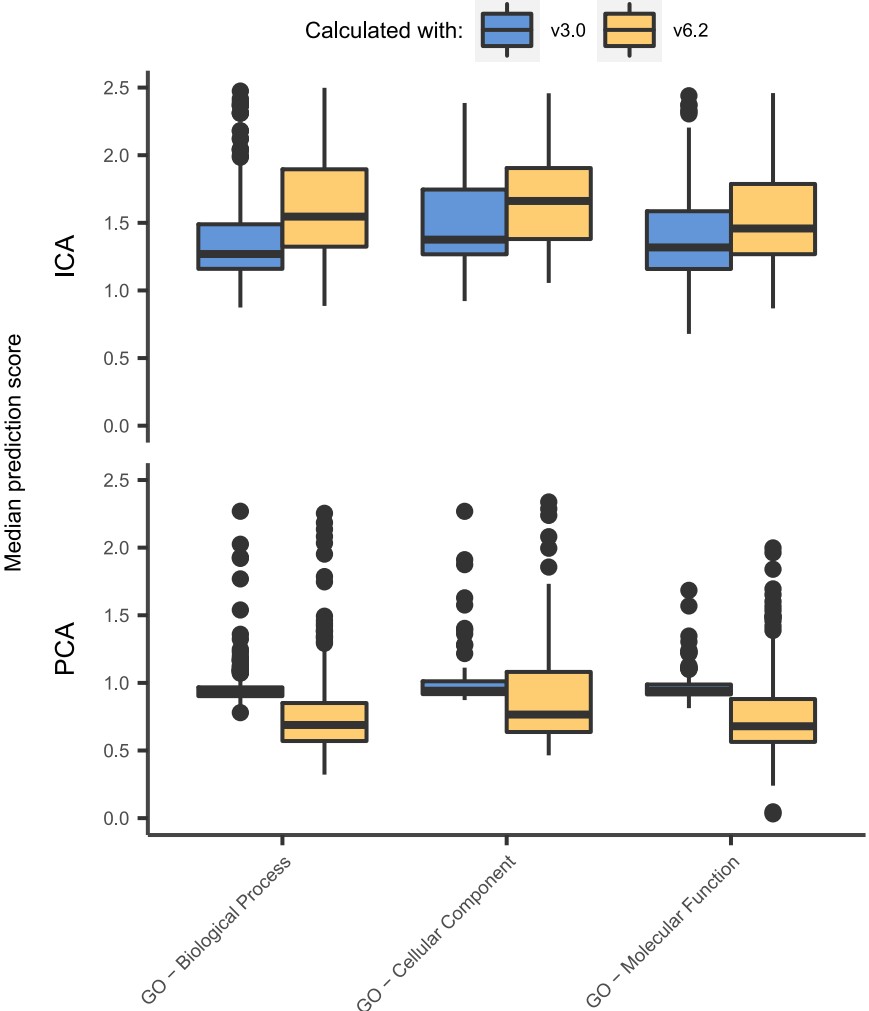

**Fig. 3 ICA-TCs outperform PCA-TCs in predicting future gene set memberships.** The predictive power of the PCA-TC- and ICA-TC-based GBA prediction strategies was further evaluated by generating predictions using a previous version of three Gene Ontology gene set collections (Molecular Signatures Database C5 v3.0). Boxplots show the prediction scores for every gene set using only genes that were added to the gene set between v3.0 (blue) and v6.2 (yellow). Prediction scores of this subset of genes degrade when using the newer version of the gene set (area under the curves (AUCs) calculated from two-sided Mann–Whitney $U$ test ranging from 0.19 to 0.34) when generating predictions with the PCA-TC- but not when using the ICA-TC-based method (AUCs ranging from 0.65 to 0.71). Only gene sets for which new members were included are depicted. Hinges of boxes represent second and third quartiles and whiskers extend by half that interquartile range. Center of box corresponds to median.

was observed between the cluster transcriptional similarity and cluster predictability scores of the 173 clusters (Fig. 5). This confirms that our method provides higher prediction scores for the biological processes that are well represented in a gene set and involves genes that are highly co-regulated. However, some clusters were observed with a low predictability score and high transcriptional similarity score. These clusters may represent co-regulated biological processes that are underrepresented or completely missing in current gene set collections. In general, the member genes of these clusters tended to be of unknown functionality, such as open-reading frames (ORFs) or uncharacterized loci (LOC). In addition, a pan-collection multifunctionality score was calculated using the gene set frequencies of all 16 gene set collections for each gene. The low multifunctionality scores of the genes belonging to these clusters reflect their low frequency of gene set memberships across all 16 gene set collections. Taken together, these results indicate that patterns of co-regulation can point towards sets of genes that participate in the same biological process even when that biological process is not well characterized in current gene set collections.

**GENETICA predicts co-functionality of uncharacterized genes.** Genes that encode uncharacterized proteins are referred to as chromosome ORFs (Corfs) or uncharacterized LOCs. Compared to the PCA-TC-based method, the ICA-TC-based method generated higher prediction scores for both Corfs and LOCs with the Hallmark gene set collection (AUC calculated with two-sided Mann–Whitney $U$ test 0.76; Fig. 6a). Hierarchical clustering of the ICA-TC-based prediction scores of 835 Corfs and LOCs based on the Hallmark gene set collection resulted in 116 clusters when using a height cutoff of 0.8. Cluster 17 was captured from 28 Corfs and LOCs that were predicted to be members of the DNA repair Hallmark gene set (Fig. 6b). Out of the 28 uncharacterized genes in this cluster, seven have been recently implicated in DNA repair based on experimental work (Fig. 6b and Supplementary Data 4). Cluster 84 consisted of Corfs predicted to be involved in estrogen receptor signaling and response (Fig. 6b). In this cluster, *C6orf141* has recently been suggested to be estrogen receptor alpha regulated in breast cancer cells[15] (Supplementary Data 4). These results show that GENETICA can predict the functionality of uncharacterized genes.

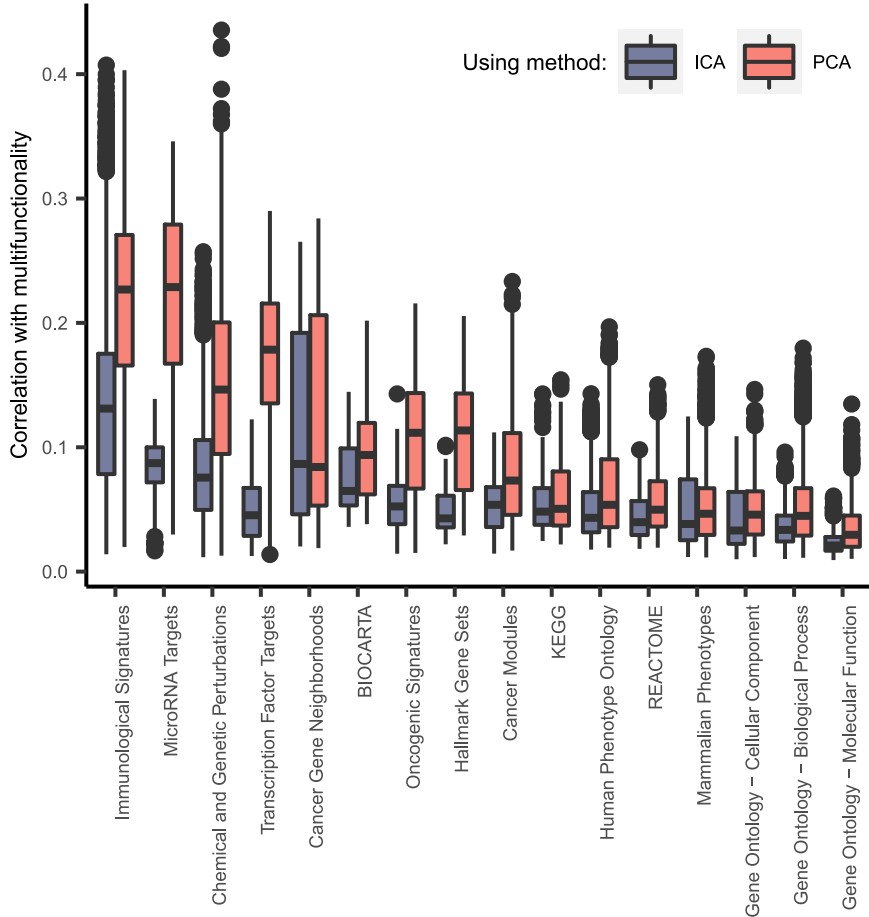

**Fig. 4 Correlation between prediction scores and gene multifunctionality.** Boxplot of distance correlation between gene set prediction scores of genes ($n = 19{,}635$) and the gene set collection multifunctionality score (y-axis) for each gene set collection (x-axis). Multifunctionality scores are compared against ICA-TC-based (blue) or PCA-TC-based (red) predictions. The magnitude of correlation varies across gene set collections and between gene sets in a collection. The associations between the degree of multifunctionality and the prediction scores were lower for the ICA-TC-based method than the PCA-TC-based method (area under the curves (AUCs) calculated from two-sided Mann–Whitney $U$ test range 0.03–0.47). Hinges of boxes represent second and third quartiles and whiskers extend by half that interquartile range. Center of box corresponds to median.

**GENETICA shows co-functionality in essentiality screen hits.** We developed a visualization tool capable of constructing networks of genes based on predicted co-functionality in one gene set collection (see "Methods" for details). Genes are clustered together in the co-functionality network based on how similar their prediction scores are for the gene sets in the selected collection. Co-functionality networks can be explored for an input list of genes to a maximum of 300.

As an example, we used the results of a recent study, in which multiple CRISPR-based genetic screens were performed in RPE-1 cells treated with different DNA-damaging agents[16]. Each screen uncovered genes that are essential for cell survival in the context of specific DNA damage-inducing agents. Overall, 840 genes were found essential for survival in the context of at least one DNA-damaging agent. Out of these 840 genes, only 334 genes were assigned to a DNA repair gene set at the time of the analysis. For the 506 genes that are currently not assigned to DNA repair pathways, the ICA-TC-based method generated higher prediction scores than the PCA-TC-based method for the "GO DNA REPAIR" gene set (Fig. 7).

Next, we constructed a co-functionality network with all 840 essential genes based on the GO Biological Process collection. We observed a large cluster containing 114 genes with high pairwise co-functionality ($R > 0.7$). A high prediction score ($Z$-score 12) was observed in this cluster for the DNA repair gene set

(Supplementary Fig. 3). Some genes in the cluster ($n = 32$) were not members of a DNA repair gene set in the original analysis, but are nevertheless known to participate in DNA repair. In other cases, these genes participate in biological processes that are indirectly associated with the cellular response to DNA damage and repair such as cell cycle regulation and nucleotide metabolism or by regulating components of the DNA damage response pathway. Less direct associations include the protein kinase C-interacting cousin of thioredoxin *PICOT*, which has only been recently shown to regulate the phosphorylation and activation of *CHK1*, *CHK2*, and *H2AX*[17]. The ubiquitin conjugase *UBE2S* has also been recently shown to bind to components of nonhomologous end joining[18]. Finally, some genes are still underexplored in the context of the cellular response to DNA repair (*GLMN*, *CFAP20*, *C1ORF112*, *SAMD1*), but have a high prediction score for DNA repair, which makes them candidates for further study.

In addition, we investigated three genome-wide CRISPR screens that aimed to capture genes involved in the inflammatory response to bacterial lipopolysaccharide (LPS)[19], lysosomal accumulation[20], and viral entry[21].

A co-functionality network was built using all 111 hit genes of the response to bacterial LPS CRISPR screen performed on dendritic cells (Fig. 8a)[19]. Two clusters were observed. The first cluster showed high prediction scores for Immunological

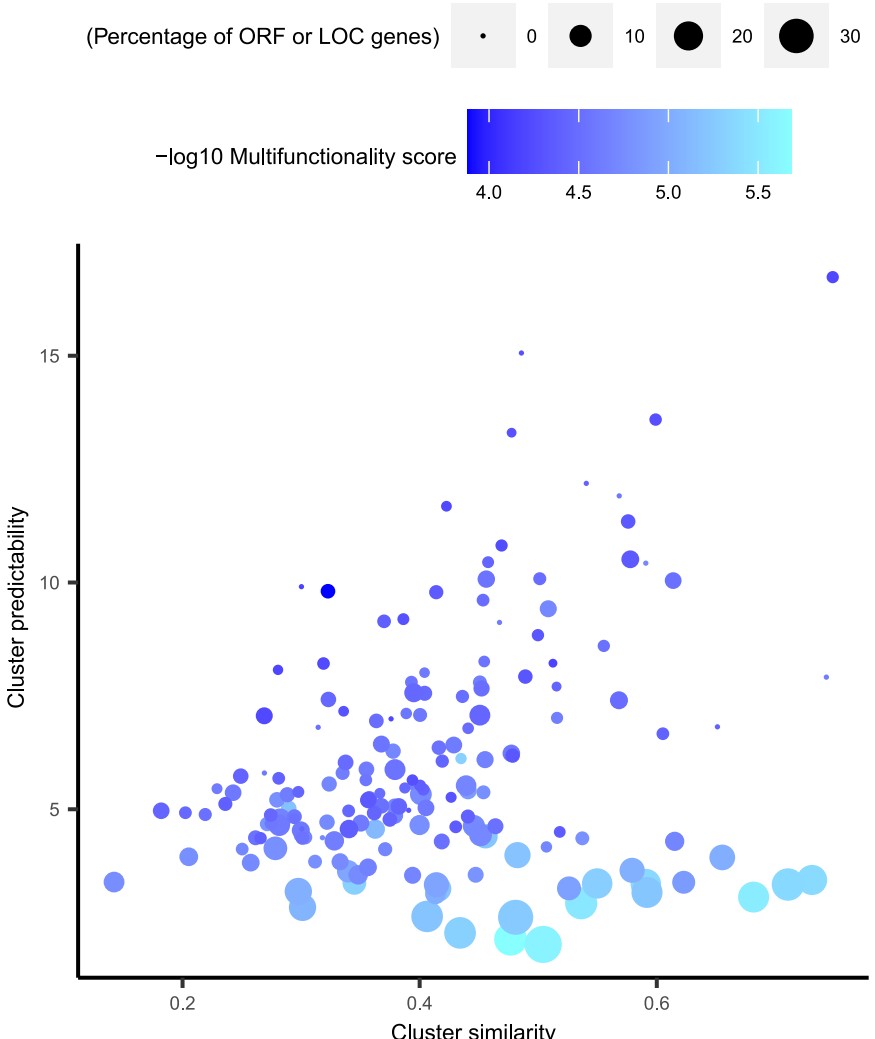

**Fig. 5 Relationship between the degree of transcriptional similarity and predictability among clusters of co-regulated genes.** Scatterplot of cluster transcriptional barcode similarity (x-axis) and cluster predictability score (y-axis) for cluster groups (n = 173). The cluster transcriptional similarity is defined as the median pairwise distance between every gene in the cluster as defined in the distance correlation matrix. A median of the maximum prediction score of each gene in the cluster was calculated to represent the cluster predictability score. Multifunctionality is depicted with a gradient from high (dark blue) to low (cyan). Clusters with high similarity but low predictability scores are composed of more uncharacterized genes (ORFs and LOCs) than other clusters. In addition, the multifunctionality scores of the genes belonging to these clusters are low (cyan), reflecting low frequency of gene set memberships.

Signatures gene sets defining the response to TREM-1, a receptor involved in amplifying the cellular response to LPS, and gene sets defining the response to *Francisella tularensis*, a pathogen that produces LPS (Fig. 8a). In contrast, the second cluster showed a high prediction score for an Immunological Signatures gene set defining reactive oxygen species-induced genes within dendritic cells (Fig. 8a). These results show the possibility to use a co-functionality network to identify distinct biological processes that are related to the phenotype under investigation in a CRISPR screen.

In the genome-wide CRISPR-knockout screen identifying 16 genes essential for lysosomal integrity, a network was built with GO Biological Process and KEGG gene set collections[20]. The ICA-TC-based prediction scores of the 16 genes for KEGG lysosome gene set ranged from 0.8 to 2.6, indicating that the ICA-TC method did not confidently predict these genes to play a role in lysosome-related processes (Fig. 8b). Reassuringly, the ICA-TC-based method still generated higher prediction scores than the PCA-TC method for the KEGG lysosome gene set. This

suggests that this particular CRISPR screen identified genes that have unique expression patterns that are incompatible within a GBA strategy.

The third genome-wide screen, studying genes involved in Ebola virus infection prioritized 13 genes[21]. Within the original publication, *GNPTAB*, a gene encoding a protein normally involved in the production of mannose-6-phosphate, was subsequently validated. Concordantly, *GNPTAB* showed the highest ICA-TC-based prediction score (prediction score = 6.4) for negative regulation of viral transcription by the host in the GO biological Process gene set collection (Fig. 8c). The other 12 genes showed prediction scores ranging from 0.7 to 2.6 for the respective gene set, demonstrating how ICA-TC-based prediction scores can be used to prioritize genes for further validation.

**Using RNA-seq data improves only ICA-TC-based predictions.** RNA-seq experiments measure more genes and have a higher dynamic range of measurement than mRNA microarray

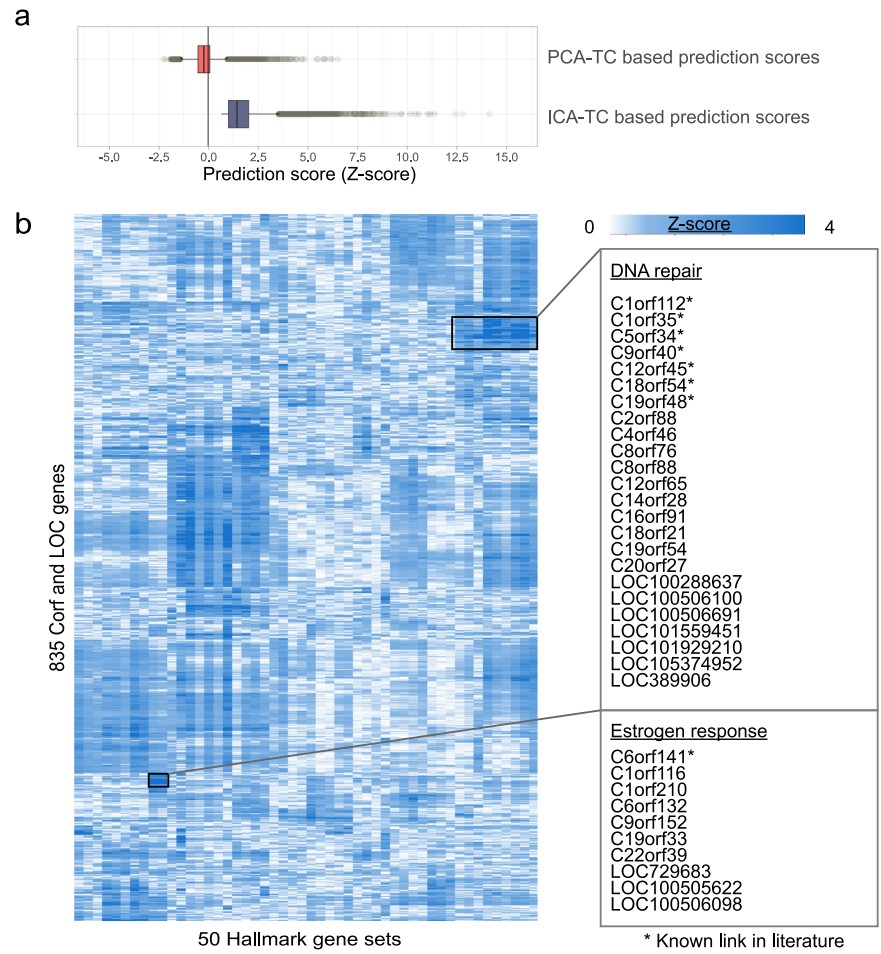

**Fig. 6 Predicted functions for uncharacterized genes using the Hallmark collection. a** Boxplot of ICA-TC-based (blue) and PCA-TC-based prediction scores (red) of Corf and LOC genes ($n = 835$) for the Hallmark gene set collection (*x*-axis). The vertical lines of boxes correspond to the median prediction scores; hinges correspond to second and third quartiles of the data; whiskers extend up to 1.5 times the interquartile range; outliers are plotted as dots. Center of box corresponds to median. **b** Heatmap depicting hierarchical clustering of Corfs and LOC genes ($n = 835$) based on the ICA-TC-based prediction scores for the Hallmark gene set collection. Clusters were obtained with a height cutoff of 0.8, resulting in 116 clusters with a median size of five members. Insets depict two clusters containing Corfs and LOC genes predicted to be involved in DNA repair, and estrogen receptor response. Genes marked with an asterisk have been recently implicated in the predicted biological processes based on experimental work.

experiments. However, the amount and diversity of publicly available microarray experiments still surpasses publicly available RNA-seq experiments. To evaluate whether RNA-seq profiles may improve upon the prediction scores generated using microarray data, the ICA-TC and PCA-TC-based methods were applied to a large set of publicly available RNA-seq samples ($n = 29,138$) (See Supplementary Notes). Using RNA-seq data did not improve the average prediction scores in most gene set collections when applying the PCA-TC-based method (AUC range: 0.12–0.41, *p* values $< 2.6 \times 10^{-4}$)(Fig. 9a). By contrast, average prediction scores improved in most gene set collections when applying the ICA-TC-based method (AUC range: 0.54–0.95, *p* values $< 2.04 \times 10^{-2}$) (Fig. 9b). The PCA-TC method had comparable multifunctionality associations when using RNA-seq and microarray input data for most gene set collections and improved in the case of the Oncogenic Signatures collection (AUC 0.21, *p* value $7.08 \times 10^{-22}$) (Fig. 9c). Predictions obtained using the ICA-TC-based method had a lower association to multifunctionality for the microRNA Targets gene set collection when using RNA-seq data as input in comparison with microarray data (AUC 0.12, *p* value $5.1 \times 10^{-40}$) (Fig. 9d). Predictions obtained using the ICA-TC-based method had a higher association to multifunctionality for the Immunological signatures gene

set collection when using RNA-seq data as input (AUC 0.59, *p* value $6.6 \times 10^{-58}$). The subset of gene set median prediction scores improved by microarray data in comparison to RNA-seq data when using ICA-TC-based method was 14.83%. These results show that the ICA-TC-based method can leverage RNA-seq profiles to improve the predictions in some gene set collections. The prediction scores based on the RNA-seq dataset have also been made available at http://genetica-network.com.

## Discussion

In this study, we utilized three large expression profile datasets generated with RNA-seq and microarray technologies, and 16 gene set collections in a GBA strategy using c-ICA to predict functional annotations. We show that our ICA-TC-based GBA strategy outperforms a currently successful method, which uses PCA-TCs in a GBA strategy. In comparison to PCA-TC-, our ICA-TC-based method (1) provides more confident functionality predictions for known and unknown gene-to-gene set combinations, (2) provides improved predictions when new knowledge is added to gene set collections, and (3) is less biased by gene multifunctionality.

Many genes do not yet have a defined function. Therefore, gene function prediction methods remain an important tool for

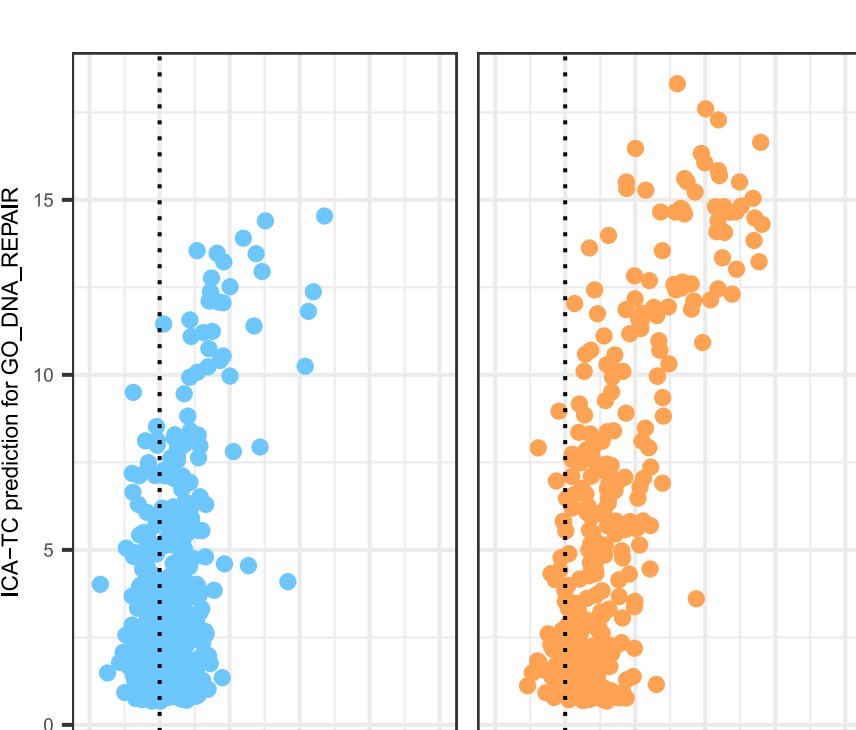

**Fig. 7 Predictions scores for genes found to be essential for DNA repair in a CRISPR-based screen.** Both the PCA-TC and ICA-TC-based method predictions for Gene Ontology Biological Process gene set GO DNA REPAIR are shown. Genes that belong (orange) or do not belong (blue) to the GO DNA repair gene set are plotted in separate panels. For the case of genes that are not members of a DNA repair gene set, the scatterplot indicates that ICA-TC-based predictions are higher, while PCA-TC-based predictions remain close to zero and are sometimes negative.

researchers. The critical assessment of functional annotation is a famous benchmark of methods that use protein sequence data to predict gene functions via gene sets[22]. Other data such as gene cross-species homology, protein–protein interaction, mRNA transcription, essentiality, and semantics can also be used to predict gene functions[5,13,23–26]. One limitation of using mRNA expression data is that some information about the gene function can only be found at the protein level using expression, interaction, or sequence data. One advantage of mRNA expression data is the greater amount of publicly available profiles in comparison to protein experiment data.

New methods such as embeddings and neural networks can identify complex relationships from protein and mRNA expression data, which serve to improve gene function predictions.

For example, autoencoders are a neural network architecture currently used to infer a latent representation of the mRNA transcription patterns obtained from both bulk and single-cell samples[27,28]. This representation can be used as a regulatory barcode to generate gene function prediction scores within a GBA strategy. Convolutional neural networks and deep neural networks have also been used to directly improve gene function prediction from the protein sequences[29,30]. The recently developed embedding technique Uniform Manifold Approximation Projection has also been utilized to predict novel protein interactions when processing mRNA expression of different gene knockout experiments[31].

In our framework, the prediction score is a $Z$-score obtained from a $p$ value. This $p$ value represents the significance level for the association between the transcriptional regulatory barcode of

an individual gene and the average barcode of a gene set. We do not assume a shape for the underlying null distribution to calculate the parametric $p$ values due to several reasons. The number of genes used to generate an average transcriptional regulatory barcode differs between gene sets (range 10–500). In addition, the association statistics used in the PCA-TC and ICA-TC-based method also differ (Pearson versus distance correlation). Finally, in the ICA-TC-based method, the average transcriptional regulatory barcode is generated by taking the mean vector of MM weights of member genes. The PCA-TC-based method instead calculates a vector of $T$ statistics resulting from the per TC Welch $T$ tests comparing member genes against nonmember genes[5]. Therefore, we utilized a permutation strategy to calculate a null distribution for each gene to gene set combination, which was then applied to calculate the $p$ values. This enabled the generation of comparable prediction scores ($Z$-scores), which allow the comparison between our ICA-TC-based method and the PCA-TC-based method that is used by DEPICT and GADO[5,13].

Prediction scores obtained with the ICA-TC-based method suggest that the majority of genes participate to a small degree in most biological processes. This observation is also in line with a recent report that shows how complex traits are associated with every gene in the genome to a small degree in the context of genome-wide association studies[32]. The PCA-TC-based method did not generate similarly high prediction scores for all genes. This suggests that the ICA-TC-based method is able to use more information than the PCA-TC method from the same input expression dataset.

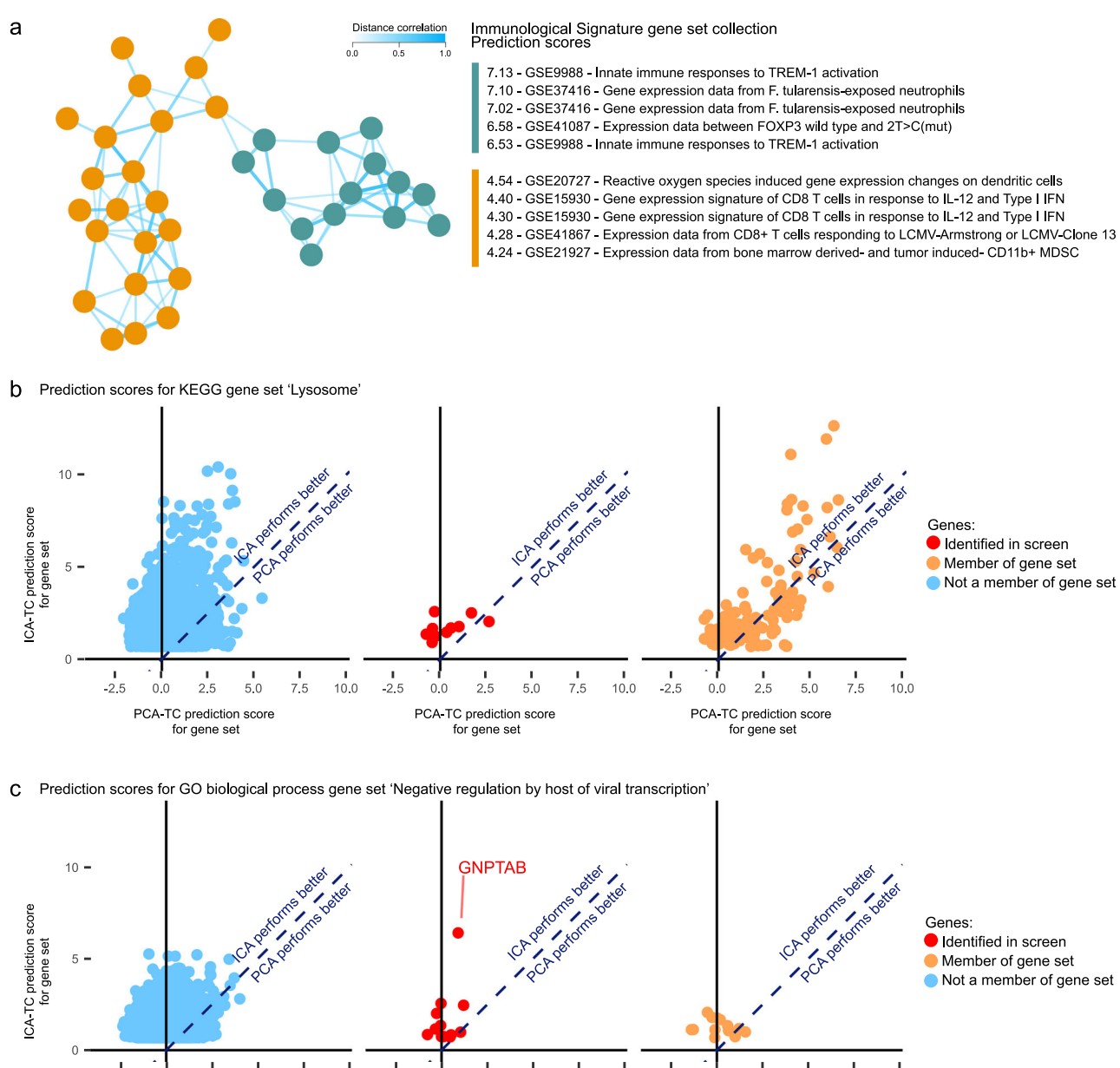

**Fig. 8 GENETICA predictions for other CRISPR-based screens. a** Co-functionality network generated with 111 genes prioritized in a CRISPR screen investigating the inflammatory response to bacterial lipopolysaccharide. Two clusters were formed that showed enrichment for predicted involvement in an inflammatory response (right cluster) and oxidative stress response in dendritic cells (left cluster). Predictions in this co-functionality network were based on the Immunological Signatures gene set collection. Only genes with a co-functionality above a threshold of $R > 0.5$ are shown in the network. **b** Scatterplots of ICA-TC-based (y-axis) and PCA-TC-based (x-axis) prediction scores of all genes for the KEGG lysosome gene set. The orange- and blue-colored dots designate the membership of genes to the respective KEGG lysosome gene set, and red dots designate genes identified in the CRISPR screen mentioned in the text. **c** Scatterplots of ICA-TC-based (y-axis) and PCA-TC-based (x-axis) prediction scores of all genes for the GO Biological Process gene set Negative regulation of viral transcription by the host. The orange and blue color dots designate the membership of genes to the respective GO Biological Process gene set, and red dots designate genes identified in the CRISPR screen mentioned in the text; the dot corresponding to the *GNPTAB* gene is highlighted.

Predicted gene functions can be used to find commonalities in the functional annotations of a group of genes without the bias to well-studied gene sets that can occur with conventional GSEA. These predictions can also enable the prioritization of genes that may be more likely to participate in a biological process from a list of candidate genes obtained as the result of an experiment. Finally, our unbiased clustering could be used to direct the future updates of gene set collections and the efforts for functional characterization of understudied co-regulated genes.

## Methods
**Data acquisition**. We downloaded the following 14 gene set collections from the Broad Institute Molecular Signatures Database v6.2 (Hallmark, Chemical and Genetic Perturbations, BIOCARTA, KEGG, REACTOME, microRNA Targets,

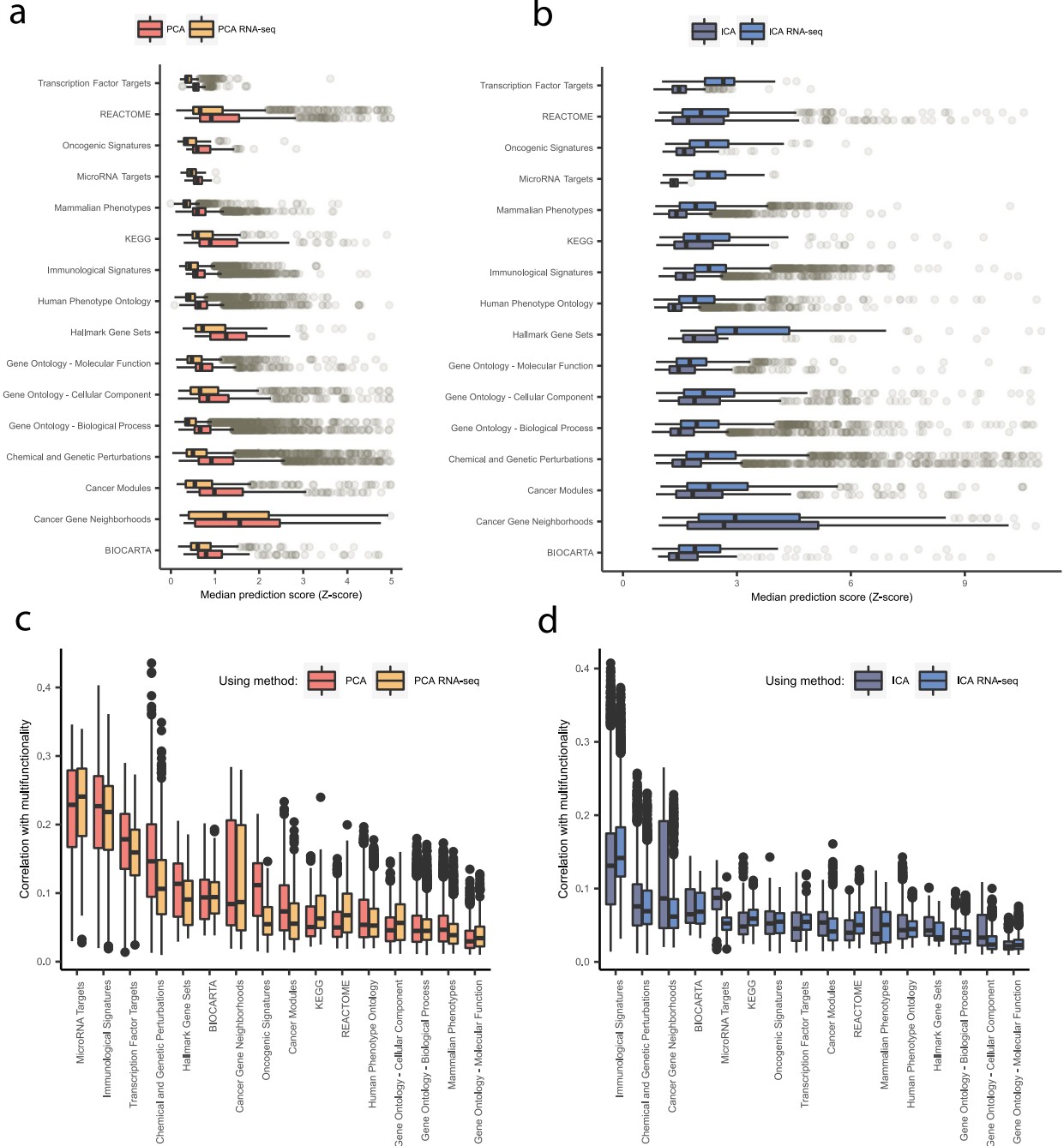

**Fig. 9 Median prediction scores and multifunctionality association for RNA-seq-based predictions. a**, **b** Boxplot of median prediction scores (*x*-axis) calculated by applying the PCA-TC-based (**a**) or ICA-TC-based (**b**) method to microarray (red and dark blue) and RNA-seq (yellow and blue) input data for each of the 16 gene set collections (*y*-axis). Median prediction scores are calculated separately for each gene set using both RNA-seq and microarray input datasets for member genes. Prediction scores of PCA-TC-based method tend to be higher when using the microarray input dataset. Prediction scores of ICA-TC-based method tend to be higher when using the RNA-seq input dataset. **c**, **d** Boxplot of distance correlation between PCA-TC-based (**c**) or ICA-TC-based (**d**) gene set prediction scores and the gene set collection multifunctionality score (*y*-axis) for each gene set collection (*x*-axis). Predictions are calculated using microarray and RNA-seq input datasets. The magnitude of correlation varies across gene set collections and between gene sets in a collection. The PCA-TC and ICA-TC-based methods have comparable multifunctionality association when using RNA-seq and microarray input data for most gene set collections. Predictions obtained using the ICA-TC-based method have a lower association to multifunctionality for the microRNA Targets and Cancer Gene Neighborhoods gene set collections when using RNA-seq data as input. Hinges of boxes represent second and third quartiles and whiskers extend by half that interquartile range. Center of box corresponds to median.

Transcription Factor Targets, Cancer Gene Neighborhoods, Cancer Modules, Oncogenic Signatures, Immunological Signatures, GO-Cellular Component, GO-Molecular Function, GO-Biological Process). In addition, we obtained two gene set collections from other sources: The Mammalian Phenotypes and Human Phenotype Ontology. We discarded from further analysis gene sets that contained <10 or >500 genes. Publicly available raw microarray expression data (CEL files) were

collected from GEO. All available samples hybridized to Affymetrix HG-U133 Plus 2.0 (GPL570) were included (April 2018).

**Quality control and preprocessing.** To identify samples that have been uploaded to GEO multiple times, we generated an MD5 hash for each individual CEL file, after

which duplicate CEL files were removed. Aggregation of raw expression data was performed according to the robust multi-array average algorithm with RMAExpress (version 1.1.0). As a quality control measure, PCA was applied on the sample correlation matrix. The first PC nearly always discriminates a platform-specific signature. Low-quality samples that did not strongly correlate with this component were discarded (Pearson $R < 0.8$). As multiple probe sets can target a single gene on Affymetrix gene expression microarrays, we utilized The R package "jetset" (version 3.4.0) to obtain one-to-one mapping between genes and the "best" probe sets for expression data generated with Affymetrix HG-U133 Plus 2.0 platform.

**PC analysis**. We performed PCA on the covariance matrix using the princomp function (R version 3.4.3) between the genes to whiten the observed variables (gene expression patterns) and reduce the dimensionality of the input data used for the c-ICA analysis. After whitening, the dimensionality was reduced from 19,635 genes to 817 whitened variables (referred in this manuscript as PCA-TCs) that explained 90% of the original variance observed in our original input dataset.

**Consensus-independent component analysis**. The FastICA algorithm utilized the whitened variables obtained by PCA to calculate independent components using the FastICA R package (version 1.2.0). The number of expected components to extract was the same as the number of withered variables ($n = 817$). To ensure that we would capture reproducible independent components, we performed 25 runs of FastICA and only retained independent components that have an equivalent component in at least 13 runs of FastICA (Pearson $R > 0.98$). A total of 523 consensus-independent components (referred in this manuscript as ICA-TCs) met this requirement. A mixing matrix was recalculated using these 523 ICA-TCs by performing a matrix multiplication against the inverse of the input dataset.

**Gene prediction scores**. The likelihood for an individual gene to be part of a biological pathway (e.g., gene set) is described by a prediction score. This will result in a vector of $n$ prediction scores (e.g., functional likelihood vector) for each individual gene for each of the $n$ gene sets in a collection. The absolute Pearson correlation between the vectors of prediction scores of two genes represents their co-functionality. A high co-functionality correlation indicates that two individual genes have similar predicted biological functions. Prediction scores were calculated using the mixing matrix and each of the 16 gene set collections was used as input using the AnalyzerTool desktop app (version 5.0).

**Co-functionality network**. Genes are plotted as nodes and their co-functionality correlation as edges in the network visualization. Only nodes with at least one correlation higher than the threshold will be shown. Selecting a group of nodes shows their average prediction scores for gene sets of the selected gene set collection.

**Gene multifunctionality score calculation**. We calculate gene multifunctionality scores for each gene set collection as described in ref.[14]. A gene multifunctionality score is a weighted sum of the number of gene sets that have that gene as a member. The weight of each membership is corrected by the size of each gene set as follows:

$$\text{Score}(G_A) = \sum_{i|G_A \in C_i} \frac{1}{N_{\text{in}_i} \times N_{\text{out}_i}}$$

Where $G_A$ is a gene, $C_i$ is a gene set in the collection that has $G_A$ as a member, $N_{\text{in}_i}$ is the number of genes that belong to $C_i$ and $N_{\text{out}_i}$ is the complementary set of genes that do not belong to $C_i$. Distance correlations were calculated using the dcor2d function from the energy R package (version 1.7).

**Hierarchical clustering of regulatory barcodes**. Using the "hclust" function, we performed a hierarchical clustering using the "ward.D2" method over an input distance matrix generated from the c-ICA mixing matrix using dcor2d (1 − correlation) (R version 3.4.3, hclust version 3.4.3, energy 1.7). Note that the mixing matrix is an output of the c-ICA and is not generated using gene sets. Clusters were generated by cutting the dendrogram at a height ($h = 2.5$), which guaranteed the size of every cluster to be within the range 10–500 genes, which is the same size range used for gene sets. This resulted in 173 clusters of sizes 13 to 389 with a mean of 115.

Uncharacterized genes were defined as genes with the "C*orf*" or the "LOC*" glob pattern in their gene symbol. For these 835 genes, the ICA-TC-based prediction scores were selected for all 50 Hallmark gene sets. The resulting data frame was clustered hierarchically, both for genes and gene sets, using the "ward. D2" method and (1 − correlation) as the distance measurement (R version 3.4.3, hclust version 3.4.3). To obtain clusters of uncharacterized genes, the gene dendrogram resulting from clustering was cut off at a height of 0.8.

**RNA-seq-based prediction scores**. The sample subset processed in the manuscript by Deelen et al.[13] was downloaded and quality controlled. Pseudocounts were obtained using kallisto and gene-level counts were calculated using DESeq2[33,34]. After quality control, 58,433 genes and 29,138 samples remained (see Supplementary Notes). In total, 85% of the variance of this dataset was

explained by 2711 PCs, which were used for calculating PCA-TC-based prediction scores. These same 2711 PCs were used as input for the consensus ICA method that generated 1373 ICs (with credibility index of 0.5), which were used to generate the ICA-TC-based prediction scores.

**Reporting summary**. Further information on research design is available in the Nature Research Reporting Summary linked to this article.

## Data availability
Prediction score tables, and corresponding input matrices for ICA- and PCA-based prediction scores for all 16 gene set collections are available on the Support data section of www.genetica-network.com. Data associated with the main figures and mouse–human prediction score correlations are available at figshare[35]: https://doi.org/10.6084/m9.figshare.13265159.

## Code availability
The software used to generate the prediction scores (Analyzertool5) can be found at https://bitbucket.org/groupfehrmann/analyzertool/src/master/ a readme with installation and usage instructions can be found in the about section of www.genetica-network.com.

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

## Acknowledgements

This research was supported by the Netherlands Organization for Scientific Research (NWO-VENI grant 916-16025 to R.S.N.F.), the Dutch Cancer Society (RUG 2013-5960 to R.S.N.F. and RUG 2016-10034 to E.G.E.d.V.), and the European Research Council (ERC Consolidator grant 682421 to M.A.T.M.v.V.).

## Author contributions

R.S.N.F. conceived this study, R.S.N.F., C.G.U.-T. and S.L. collected and assembled the data. Data analyses were performed by C.G.U.-T., R.S.N.F., A.B. and V.C.L. C.G.U.-T., V.C.L., A.B., S.L., E.G.E.d.V., M.A.T.M.v.V. and R.S.N.F. contributed to the data interpretation, writing of the manuscript, and the final decision to submit the manuscript.

## Competing interests

The authors declare no competing interests.
