## [Peer Review File · Nature Communications]

Reviewers' Comments:

Reviewer #1:

Remarks to the Author:

In this study, Urzua-Traslavina and his colleagues improved the prediction of gene-set membership of pre-defined gene sets using ICA-TC and GBA strategy. Technically, this work is well-written with promising results, but from biological perspective, it is not well-written and it is not so informative. Actually, in this paper, the authors just used of one biological example "DNA Repair" gene-set.

I highly recommend;

1- provide more information about the previous devised methods for gene function prediction, specially with focus on machine learning approaches.

2- provide a list of gene-sets that the gene-set membership has been improved by the developed approach and show that, whether all of the gene-sets have been improved? if not, explain why?

3- Did you also work with gene expression time-series data? If you excluded them, please explain why? I suppose they would be better data for the evaluation of the method.

Reviewer #3:

Remarks to the Author:

Summary:

The manuscript by Urzua-Traslavina et al. is mostly well-written (it misses some important parts as mentioned below) and builds to a large extent on a previous highly-cited study by the senior author Dr. Fehrmann (Nature Genetics 2015, PMID 25581432). Here, Urzua-Traslavina et al. use 106,462 samples from the Affymetrix HG-U133 Plus 2.0 platform to construct 523 transcriptional components (TCs) using a consensus independent component analysis (ICA)-based strategy. Based on the cICA mixing matrix they build "average transcriptional barcodes" for each of 23,372 genesets from the Human Phenotype Ontology, the Mouse Genome Informatics database and the Molecular Signatures database. They then correlate genes' transcriptional barcodes with these genesets' transcriptional barcodes to produce predictions on whether a given gene is associated with a given geneset. They make the predictions publicly available. They find that their ICA-TC framework outperforms a principal component analysis-TC-based framework. In general, their ICA-TC framework is highly valuable for more agonistic analyses of genes and their functions.

Major comments:

The focus on Affymetrix gene expression technology seems a bit outdated. Because more than 80% of all GWAS associations typically fall outside coding regions and because non-coding genes are understudied compared to protein-coding genes, it would be very useful if the ICA-TCs were computed based on large-scale RNA-seq data instead of microarray data. RNA-seq data could provide predictions for the much larger set of GENCODE genes, results that supposedly would be extremely valuable to better understand contributions of non-coding genes to complex traits and disease. Also, from a technical perspective, how much do their predictions increase when using state-of-the-art RNA-seq data?

Besides the narrow focus on microarray data, another concern is that lack of single-cell RNA-seq data. Can their approach be applied on single-cell RNA-seq data which is typically sparse and where less plentiful compared to bulk expression data? Would be great if the authors, at least, could include examples on how their approach works when e.g. applied on all single-cell data from some of the large multi-organ and comprehensive brain atlases. For instance, will much few samples be needed

because the data is much more specific than e.g. bulk microarray data?

There is no mentioning of species. Are these human data or a mix between human, mouse and rat samples? What are the differences in their benchmarks between species?

The authors state that "Some datasets that were not included in the article due to size constraints and the source code for the method are made available from the corresponding author on reasonable request". To better facilitate that other research groups can use their framework (besides browsing the website) the authors should provide all code (for the microarray, and should they decide to do these analysis, the RNA-seq and single-cell RNA-seq parts) and all ICA-TCs (by-species and combined TCs) via a publicly accessible website (instead of "upon request").

Minor comments:

P. 3: Spell out MSigDB

P. 4: Mention examples on what it means that a "a gene is likely to play a role in that gene set".

P. 5: Table not properly described. For instance, it is not clearly stated that GENETICA is the method proposed by the authors.

I would like to conclude by congratulating the authors on this important contribution and thank them for their work which will be highly relevant and useful to the community.

My best

Tune H Pers

Urzua-Traslavina *et al.*: Improving gene function predictions using independent transcriptional components (NCOMMS-20-18207)

We would like to thank the reviewers for their helpful comments. These constructive remarks have enabled us to improve our manuscript. Below, please find a point-by-point rebuttal.

Referee 1

Remark #1:

“... from biological perspective, it is not well-written, and it is not so informative. Actually, in this paper, the authors just used of one biological example ”DNA Repair” gene-set”

Reply: We agree with this reviewer that the manuscript lacked the biological perspective of our findings. To expand the biological perspective of the manuscript, we added several additional analyses.

Firstly, we explore the predicted functional landscape according to the Hallmark gene set collection for a subset of genes, particularly, chromosome open reading frame (Corf) genes and uncharacterized loci (LOC) genes. Hierarchical clustering of the Hallmark gene-set collection prediction scores of uncharacterized genes revealed several clusters with distinct predicted biological processes, such as response to estrogen receptor activity and DNA repair (Figure 6). We identified several uncharacterized genes for which the biological function was experimentally demonstrated only recently and matched our predictions (Supplementary Data 4).

We added the following sentences to our Results section on Page 9, Line 23 :

“Genes that encode uncharacterized proteins are referred to as chromosome open reading frames (Corfs) or uncharacterized loci (LOCs). Compared to the PCA-TC based method, the ICA-TC based method generated higher prediction scores for both Corfs and LOCs with the Hallmark gene-set collection (Area under the curve (AUC) calculated with Mann–Whitney U test 0.76; Figure 6A). Hierarchical clustering of the ICA-TC based prediction scores of 835 Corfs and LOCs based on the Hallmark gene-set collection resulted in 116 clusters when using a height cutoff of 0.8. Cluster 17 was captured 28 Corfs and LOCs that were predicted to be members of the DNA repair Hallmark gene set (Figure 6B). Out of the 28 uncharacterized genes in this cluster, seven have been recently implicated in DNA repair based on experimental work (Figure 6B, Supplementary Data 4). Cluster 84 consisted of Corfs predicted to be involved in estrogen receptor signaling and response (Figure 6B). In this cluster, *C6orf141* has recently been suggested to be estrogen receptor alpha regulated in breast cancer cells [1] (Supplementary Data 4). These results show that GENETICA can predict the functionality of uncharacterized genes. ”

We added these analyses in a new Figure 6:

We added the following legend for Figure 6:

“Predicted functions for uncharacterized genes using the Hallmark collection.

A) Boxplot of ICA-TC based (blue) and PCA-TC based prediction scores (red) of Corf and LOC genes, for the Hallmark gene set collection (x-axis). The vertical lines of boxes correspond to the median prediction scores; hinges correspond to first and third quartiles of the data; whiskers extend up to 1.5 times the interquartile range; outliers are plotted as dots.

B) Heatmap depicting hierarchical clustering of Corfs and LOC genes based on the ICA-TC based prediction scores for the Hallmark gene set collection. Clusters were obtained with a height cutoff of 0.8 resulting in 116 clusters with a median size of 5 members. Insets depict two clusters containing Corfs and LOC genes predicted to be involved in DNA repair, and estrogen receptor response. Genes marked with an asterisk have been recently implicated in the predicted biological processes based on experimental work. ”

Secondly, we predicted the function of genes prioritized with CRISPR based genetic screens in three additional biological contexts (Figure 8), namely the inflammatory response to bacterial lipopolysaccharide, viral entry, and lysosome accumulation.

We added the following sentences to our Results section on Page 11, Line 11 :

“Additionally, we investigated three genome-wide CRISPR screens that aimed to capture genes involved in the inflammatory response to bacterial lipopolysaccharide [2], lysosomal accumulation [3], and viral entry [4].

A co-functionality network was built using all 111 hit genes of the response to bacterial lipopolysaccharide (LPS) CRISPR screen performed on dendritic cells (Figure 8A) [2]. Two clusters were observed. The first cluster showed high prediction scores for Immunological Signatures gene sets defining the response to TREM-1, a receptor involved in amplifying the cellular response to LPS, and gene sets defining the response to *F. tularensis*, a pathogen that produces LPS (Figure 8A). In contrast, the second cluster showed a high prediction score for an Immunological Signatures gene set defining reactive oxygen species-induced genes within dendritic cells (Figure 8A). These results show the possibility to use a co-functionality network to identify distinct biological processes that are related to the phenotype under investigation in a CRISPR screen.

In the genome-wide CRISPR–knockout screen identifying 16 genes essential for lysosomal integrity, a network was built with GO Biological Process and KEGG gene set collections [3]. The ICA-TC based prediction scores of the 16 genes for KEGG lysosome gene set ranged from 0.8 to 2.6, indicating that the ICA-TC method did not confidently predict these genes

to play a role in lysosome-related processes (Figure 8B). Reassuringly, the ICA-TC based method still generated higher prediction scores than the PCA-TC method for the KEGG lysosome gene set. This suggests that this particular CRISPR screen identified genes that either have unique expression patterns that are incompatible within a guilt-by-association-strategy.

The third genome-wide screen, studying genes involved in Ebola virus infection prioritized 13 genes [4]. Within the original publication, *GNPTAB*, a gene encoding a protein normally involved in the production of mannose-6-phosphate, was subsequently validated. Concordantly, *GNPTAB* showed the highest ICA-TC-based prediction score (Prediction score = 6.4) for negative regulation of viral transcription by the host in the GO biological Process gene set collection (Figure 8C). The other 12 genes showed prediction scores ranging from 0.7 to 2.6 for the respective gene set, demonstrating how ICA-TC-based prediction scores can be used to prioritize genes for further validation. ”

These data are now added in a new Figure 8:

We added the following legend for Figure 8:

“GENETICA predictions for other CRISPR based screens. A) Co-functionality network generated with 111 genes prioritized in a CRISPR screen investigating the inflammatory response to bacterial lipopolysaccharide. Two clusters were formed that showed enrichment for predicted involvement in an inflammatory response (right cluster) and oxidative stress response in dendritic cells (left cluster). Predictions in this co-functionality

network were based on the Immunological Signatures gene set collection. Only genes with a co-functionality above a threshold of $R > 0.5$ are shown in the network. **B)** Scatterplots of ICA-TC based (y-axis) and PCA-TC based (x-axis) prediction scores of all genes for the KEGG lysosome gene set. The orange and blue colored dots designate the membership of genes to the respective KEGG lysosome gene set, red dots designate genes identified in the CRISPR-screen mentioned in the text. **C)** Scatterplots of ICA-TC based (y-axis) and PCA-TC based (x-axis) prediction scores of all genes for the GO Biological Process gene set Negative regulation of viral transcription by the host. The orange and blue color dots designate the membership of genes to the respective GO Biological Process gene set, red dots designate genes identified in the CRISPR screen mentioned in the text; the dot corresponding to the GNPTAB gene is highlighted. ”

Remark #2:

“...provide more information about the previously devised methods for gene function prediction, with special focus on machine learning approaches.”

Reply: We appreciate this recommendation and include a short overview of the methods of gene prediction in the discussion section of the manuscript (Page 14, Line 1).

“Many genes do not yet have a defined function. Therefore, gene function prediction methods remain an important tool for researchers. The critical assessment of functional annotation (CAFA) is a famous benchmark of methods that use protein sequence data to predict gene functions via gene sets [5]. Other data such as gene cross-species homology, protein-protein interaction, mRNA transcription, essentiality, and semantics can also be used to predict gene functions [6–11]. One limitation of using mRNA expression data is that some information about the gene function can only be found at the protein level using expression, interaction, or sequence data. One advantage of mRNA expression data is the greater amount of publicly available profiles than protein experiment data.

New methods such as embeddings and neural networks can identify complex relationships from protein and mRNA expression data which serve to improve gene function predictions.

For example, autoencoders are a neural network architecture currently used to infer a latent representation of the mRNA transcription patterns obtained from both bulk and single cell samples [12, 13]. This representation can be used as a regulatory barcode to generate gene function prediction scores within a guilt-by-association strategy. Convolutional neural networks and deep neural networks have also been used to directly improve gene function prediction from the protein sequences [14, 15]. The recently developed embedding technique Uniform Manifold Approximation Projection (UMAP) has also been utilized to predict novel protein interactions when processing mRNA expression of different gene knockout experiments [16]. Predicted protein interactions can also be used to predict function predictions by propagating the interacting partner functions. ”

Remark #3:

“provide a list of gene-sets that the gene-set membership has been improved by the developed approach and show that, whether all of the gene-sets have been improved? if not, explain why?”

Reply: To address this remark, we compared PCA-TC based median prediction scores with ICA-TC based prediction scores for all gene sets and calculated a difference (delta) Supplementary Data 1. This analysis revealed that only a minority (3.4%) of gene sets are not improved and comprise a smaller set of genes than the improved gene sets (median gene members is 23 and, 57 respectively.). Gene sets derived from one study (NIKOLSKY prefix) were overrepresented in the top 20 PCA-improved subset. These gene sets represented amplified genomic loci in breast cancer samples that were not associated with any biological process in the original publication [17].

We added the following sentences to our Results section on Page 6, Line 7 :

“To explore if a subset of gene sets is better predicted with the PCA-TC based method

the differences between gene set prediction scores of both PCA and ICA methods were calculated (delta). Gene sets with deltas higher than zero were defined as ICA-improved and below zero as PCA-improved gene sets. We observed that 811/23,413 (3.4%) gene sets belonged to the PCA-improved subset (Supplementary Fig. 1). Gene sets belonging to the PCA-improved subset had fewer members (median = 23) than the ICA-improved subset (median = 57). ”

We added Supplementary Fig. 1:

We added the following legend for Supplementary Fig. 1:

“Gene sets where the PCA-TC based method provided better median prediction scores. **A**) Scatterplot of PCA-TC (x-axis) versus ICA-TC (y-axis) based median prediction scores. The scatterplot only shows the 811/23,413 (3.4%) of gene sets for which the PCA-TC based method produced better median prediction scores in comparison to the ICA-TC based method. The size and color of dots represent the number of genes in a gene set. The inset shows the scatterplot with all 23,413 gene sets. **B**) Difference between the PCA-TC and ICA-TC based median prediction scores (y-axis) for the 811 gene sets that show higher prediction scores for the PCA-TC based method. The gene sets are ranked on the x-axis based on the difference and grouped according to gene set collections. The legend shows each gene set collection and the percentage of gene sets from that collection

that showed better median prediction scores with the PCA-TC based method ”

Remark #4:

“Did you also work with gene expression time-series data? If you excluded them, please explain why? I suppose they would be better data for the evaluation of the method.”

Reply: We did not evaluate whether the ICA-TC based method could be used to predict the mRNA profile of a later timepoint using an earlier timepoint in time-series experiments. This is because predicting temporal gene correlations is not an objective of our study. We unbiasedly collected all samples processed with the Affymetrix HG U-133 Plus 2.0 microarray platform (GPL570). These samples correspond to a wide variety of tissue and cell line experiments, of which a subset represents biological replicates measured at different timepoints. The ICA-TC based method benefits from the inclusion of a wide variety of mRNA expression profiles which are sometimes represented by the early and later timepoints of a perturbation experiment. Therefore, studies with timepoints were not purposefully excluded from the subsequent analysis.

To showcase that our analysis captured the transcriptional activity at different timepoints in time-series experiments we selected two studies from our input mRNA dataset. In the first study (GSE67684) mRNA profiles of 210 blood or bone marrow biopsies were generated before and 8 days after remission induction therapy in children with de-novo acute lymphoblastic leukemia [18]. In the second study (GSE12548) triplicate mRNA profiles of ARPE-19 cells were generated at different timepoints before and after combined treatment with TGF-beta and TNF-alpha [19]. Using UMAP to visualize the weights in the independent components corresponding to the patient samples reveals that their transcriptional profiles tend towards a common center after remission induction therapy (Peer Review Fig. 1A) [20]. Similarly, ARPE-19 cell replicates consistently showed changes in expression that shift their position in the embedded space (Peer Review Fig. 1B).

Peer Review Figure 1: UMAP embedding of two time series experiments. A) Linked scatterplot of 2D UMAP embedding coordinates of cancer patient biopsies before and after 8 days of treatment (GSE67684). UMAP was performed on the independent component weights corresponding to these samples. Color corresponds to timepoint. Transparency of point and line corresponds to Euclidean distance traveled by the biopsy transcriptional profile between timepoints. **B)** Scatterplot of 2D UMAP embedding coordinates for ARPE-19 replicates at different timepoints of TGF-TNF treatment (GSE12548). Color corresponds to timepoint.

Referee 2

We did not receive any comments from the second invited reviewer.

Referee 3

We would like to thank the reviewer for the acknowledgement that the manuscript is mostly well-written and that the reviewer finds our ICA-TC framework highly valuable for unbiased analyses of genes and their functions. We are also happy with his congratulations on this important contribution and his belief that our work will be highly relevant and useful to the community.

Remark #1:

“The focus on Affymetrix gene expression technology seems a bit outdated. More than 80% of all GWAS associations typically fall outside coding regions and non-coding genes are understudied compared to protein-coding genes. Therefore, it would be very useful if the ICA-TCs were computed based on large-scale RNA-seq data instead of microarray data. RNA-seq data could provide predictions for the much larger set of GENCODE genes, results that supposedly would be extremely valuable to better understand contributions of non-coding genes to complex traits and disease. Also, from a technical perspective, how much do their predictions increase when using state-of-the-art RNA-seq data?”

Reply: We thank the reviewer for this recommendation and completely agree that it would greatly improve the impact of our method. To address this remark we downloaded and generated gene-level counts ($n = 58,433$) for quality-controlled samples ($n = 29,138$)

provided in the GeneNetwork Assisted Diagnostic Optimization manuscript [8]. We kept all parameters and gene set collections consistent with the microarray version of our framework except the explained variance set to 85% for computational restrictions. Using RNA-seq data did not improve the average prediction scores in most gene set collections when applying the PCA-TC based method (Figure 9A). By contrast, average prediction scores improved when applying the ICA-TC based method (Figure 9B). This new version of our framework is now also available at <http://www.genetica-network.com>.

We added the following sentences to our Results section on Page , Line :

“RNA-seq experiments measure more genes and have a higher dynamic range of measurement than mRNA microarray experiments. However, the amount and diversity of publicly available microarray experiments still surpasses publicly available RNA-seq experiments. To evaluate whether RNA-seq profiles may improve upon the prediction scores generated using microarray data, the ICA-TC and PCA-TC based methods were applied to a large set of publicly available RNA-seq samples ($n = 29, 138$) (See Supplementary Notes). Using RNA-seq data did not improve the average prediction scores in most gene set collections when applying the PCA-TC based method (AUC range: 0.12 - 0.41, p-values $< 2.6 \times 10^{-4}$) (Figure 9A). By contrast, average prediction scores improved in most gene set collections when applying the ICA-TC based method (AUC range: 0.54 - 0.95, p-values $< 2.04 \times 10^{-2}$) (Figure 9B). The PCA-TC method had comparable multifunctionality association when using RNA-seq and microarray input data for most gene set collections and improved in the case of the Oncogenic Signatures collection (AUC 0.21, p-value 7.08×10^{-22}) (Figure 9C). Predictions obtained using the ICA-TC based method had a lower association to multifunctionality for the microRNA Targets gene set collection when using RNA-seq data as input in comparison with microarray data (AUC 0.12, p-value 5.1×10^{-40}) (Figure 9D). Predictions obtained using the ICA-TC based method had a higher association to multifunctionality for the Immunological signatures gene set collection when using RNA-seq data as input (AUC 0.59, p-value 6.6×10^{-58}). The subset of gene set median prediction scores improved by

microarray data in comparison to RNA-seq data when using ICA-TC based method was 14.83%. These results show that the ICA-TC based method can leverage RNA-seq profiles to improve the predictions in some gene set collections. The prediction scores based on the RNA-seq dataset have also been made available at <http://genetica-network.com>. ”

These data are now added in a new Figure 9:

We added the following legend for Figure 9:

“Median prediction scores and multifunctionality association for RNA-seq based predictions. A-B) Boxplot of median prediction scores (x-axis) calculated by applying the PCA-TC (A) or ICA-TC (B) based method to microarray and RNA-seq input

data for each of the 16 gene-set collections (y-axis). Median prediction scores are calculated separately for each gene set using both RNA-seq and microarray input datasets for member genes. Prediction scores of PCA-TC based method tend to be higher when using the microarray input dataset. Prediction scores of ICA-TC based method tend to be higher when using the RNA-seq input dataset. **C-D**) Boxplot of distance correlation between PCA-TC **(C)** or ICA-TC **(D)** based gene-set prediction scores and the gene-set collection multifunctionality score (y-axis) for each gene-set collection (x-axis). Predictions are calculated using microarray and RNA-seq input datasets. The magnitude of correlation varies across gene set collections and between gene sets in a collection. The PCA-TC and ICA-TC based methods have comparable multifunctionality association when using RNA-seq and microarray input data for most gene set collections. Predictions obtained using the ICA-TC based method have a lower association to multifunctionality for the microRNA Targets and Cancer Gene Neighborhoods gene set collections when using RNA-seq data as input. Hinges of boxes represent second and third quartiles and whiskers extend by half that interquartile range. ”

We added the following sentences to our Supplementary Notes on Page 8, Line 19 :

“We downloaded and quality controlled the sample subset processed in the manuscript by Deelen et al using the ftp link table provided in the Supplementary Notes of the original publication [8]. We generated pseudocounts using Kallisto 0.46.0 [21] specifying default parameters for paired-end data and the following additional settings for single-end data: -single -l 200 -s 20 -bias. The following two genome files were merged and used to create the Kallisto index:

```
ftp://ftp.ensembl.org/pub/release-101/fasta/homo_sapiens/cdna/Homo_sapiens.  
GRCh38.cdna.all.fa.gz  
ftp://ftp.ensembl.org/pub/release-101/fasta/homo_sapiens/ncrna/Homo_sapiens.  
GRCh38.ncrna.fa.gz
```

In total 31,395 samples were successfully downloaded. For quality control, we dropped

samples with less than 70% of pseudoaligned reads and where the read count did not match the count reported by the ftp link table, leaving us with 30,288 samples. For duplicated samples with a correlation of > 0.9999 , one was randomly selected and the others removed which left 29,138 samples for our analysis. Next, transcripts that were not located on the main chromosomes (1-22, X, Y, MT), but on scaffolds were removed, reducing the number of transcripts from 250,156 to 228,267. Another 965 transcripts with 100% identical sequences were removed to avoid a double-mapping bias while randomly keeping one transcript. Transcripts with less than one non zero counts were dropped, ending up with 227,300 transcripts.

The transcript counts per sample were summed up to gene-level counts for each sample, which yielded 59,030 genes. Duplicated gene names occurred in 14 instances and were removed while keeping the newest version of each. At last, we removed genes expressed in less than 1% of the samples, ending up with a final gene-count matrix of 58,433 genes and 29,138 samples. Finally, the gene counts were normalized using size factors and variance stabilizing transformation from DESeq2 1.26.0 [22].

To map Ensembl gene IDs to entrez IDs from NCBI, a table containing both was downloaded from HGNC (<https://pubmed.ncbi.nlm.nih.gov/30304474/>) on 19.10.2020. Missing NCBI IDs were filled with the newest non-curated ones, supplied directly by NCBI, ending up with 38,381 genes identified with Ensembl gene IDs to entrez IDs mapping. ”

Remark #2:

“Besides the narrow focus on microarray data, another concern is that lack of single-cell RNA-seq data. Can their approach be applied on single-cell RNA-seq data which is typically sparse and where less plentiful compared to bulk expression data?. Would be great if the authors, at least, could include examples on how their approach works when e.g. applied on all single-cell data from some of the large multi-organ and comprehensive brain atlases. For instance, will much few samples be needed because the data is much more specific than e.g. bulk microarray data?”

Reply: We thank the reviewer for this observation. We agree that single cell data captures a higher resolution of the mRNA expression of single cells and could potentially improve gene predictions. However, single-cell profiles are currently very sparse with sometimes only 10% of genes with non-zero measurements. This complicates the application of our method since more samples are required to have enough non-zero measurements to calculate a complete gene-by-gene covariance matrix. Additionally, we have not yet determined which cross-study normalization procedure is the most appropriate preprocessing for our method before the calculating the covariance matrix.

We believe that the biopsy variety of publicly available experiments is still higher in bulk mRNA data. For example, we explored panglaoDB and extracted all human experiments performed using 10x chromium technology ($n = 239$, cells = 397,071) [23]. The majority of experiments were performed on lymphoid tissues, blood, prostate, testis, brain, kidney and liver. In contrast, publicly available bulk mRNA experiments are obtained from healthy, diseased and cancerous complex biopsies of many more tissues in addition to cell line perturbation experiments. Since our method relies on the variety of mRNA states to extract reliable transcriptional components with ICA, this hampers the attractiveness of currently available single cell mRNA profiles for gene function prediction. Still, we cannot rule out that single cell could enable better function predictions when more profiles and future technologies become available.

Remark #3:

“There is no mentioning of species. Are these human data or a mix between human, mouse and rat samples? What are the differences in their benchmarks between species?”

Reply: In our original analysis we only used samples generated with the Affymetrix HU-113 Plus 2.0 platform ($n = 106,462$), which is a human microarray. We did calculate predictions for a mouse gene set collection (Mammalian Phenotypes) using the human ortholog mapped gene sets provided at the same resource (See Supplementary Notes). To evaluate whether gene to gene set predictions differs across species we generated predictions

scores using the Affymetrix Mouse Genome 430 2.0 microarray platform (GEO accession identifier: GPL1261). A subset of 10,974/14,589 (75.2%) mouse genes have a correlation > 0.3 to their human orthologs based on their prediction scores for at least one of the gene set collections (Supplementary Fig. 2). The mouse prediction scores are available at genetICA-network.com.

We added the following sentences to our Results section on Page 8, Line 6 :

“In addition, ICA-TC based gene function predictions were generated for mouse microarray samples (GPL1261 platform; $n = 25,585$; *genes* = 18,425), for 16 gene set collections ($n = 23,128$). We correlated the prediction scores between the unique ortholog mapping genes ($n = 14,589$) between human and mouse microarray datasets for every matching gene set collection to investigate the concordance between prediction scores (Supplementary Fig. 2). A subset of 10,974/14,589 (75.2%) mouse genes have a correlation > 0.3 to their human orthologs based on their prediction scores for at least one of the gene set collections. GO Cellular Component and Cancer Modules gene set collections showed the highest median Spearman correlations (0.155 and 0.154, respectively). The low correlating gene predictions across species could be explained by the smaller number of samples ($n = 25,585$) and the more limited sample heterogeneity available in the mouse dataset compared to the human microarray dataset ($n = 106,462$). These results show that many genes may be similarly regulated in both species. The mouse prediction scores have been made available at <http://www.genetica-network.com>.

”

These data are added in Supplementary Fig. 2:

We added the following legend for Supplementary Fig. 2:

“**Cross-species prediction score correlations.** Histograms showing the distributions of Spearman correlation coefficients between the ICA-TC and PCA-TC based prediction scores of every mouse-human gene ortholog pair for 16 gene set collections. Median Spearman correlation between ortholog prediction scores ranged between 0 and 0.2 for every gene set collection. The number of gene set perspective comparisons performed for each collection is depicted in the y axis text. ”

We added the following sentences to our Supplementary Notes on Page 7, Line 2 :

“We downloaded mRNA expression profiles of all mouse samples available for the Affymetrix Mouse Genome 430 2.0 microarray platform in their unprocessed CEL format (GPL1261). To remove potential duplicates MD5 digests were calculated for all CEL files and one replicate was retained. All CEL files were then transformed to gene-level normalized expression values using the RMAExpress algorithm (1.20.0). To remove potential bad quality experiments principal component analysis (PCA) was applied to the normalized expression matrix. In microarray expression data principal component one (PC1) captures a platform-specific signature that is shared by all samples. Samples that do not share this signature may represent bad quality experiments. All samples with a PC1 lower than the 80th percentile were dropped ending up with 56,657 samples.

Using the same methodology used for the human microarray we subsequently applied PCA again to whiten the normalized expression matrix and observed that the number of components needed to capture 90% of the dataset variance was 753. We then applied independent component analysis to the whitened matrix targetting the acquisition of 753 components on 25 different runs. We retained sources present in at least 13 runs (Credibility index = 0.5) ending up with 614 consensus mouse ICA-TCs. As multiple probesets can target a single gene on Affymetrix gene expression microarrays we utilized The R package Jetset (version 3.4.0) to obtain one-to-one mapping between genes and the highest quality probesets for expression data generated with the Affymetrix Mouse Genome 430 2.0 microarray platform [24].

Creating mouse gene set collections based on Molecular Signal Database human gene set collections

To generate gene sets in gmt-file format based on the human gene set collections from Molecular Signatures Database v6.2, human gene entrez IDs were translated to mouse gene entrez IDs using the annotations provided in the msigdb R package (version 7.2.1). The msigdb package was built manually using the “msigdb-prepare.R script”, that was available at <https://github.com/igordot/msigdb/>, to be able to collect Molecular Signatures Database v6.2 gene sets: line 14 was modified (“msigdb version = ”6.2”). For Human Phenotype

Ontology and Mammalian Phenotypes gene set collections human entrez IDs were translated to mouse entrez IDs by using the BioMart datasets (ensembl) available through the biomaRt R package (version 2.45.9).

”

Remark #4:

“The authors state that: Some datasets that were not included in the article due to size constraints and the source code for the method are made available from the corresponding author on reasonable request. To better facilitate that other research groups can use their framework (besides browsing the website) the authors should provide all code (for the microarray, and should they decide to do these analysis, the RNA-seq and single-cell RNA-seq parts) and all ICA-TCs (by-species and combined TCs) via a publicly accessible website (instead of upon request)”

Reply: To address this remark, we deposited the prediction scores used in this manuscript and input matrices in the download section at www.genetica-network.com and all figure data at a Figshare repository.

We amended the data availability statement in Page 19, Line 1 :

“Prediction score tables, and corresponding input matrices for ICA and PCA based prediction scores for all 16 gene set collections are available on the download section of www.genetica-network.com Data associated with the main figures and mouse-human prediction score correlations are available at Figshare: <https://doi.org/10.6084/m9.figshare.13265159>”

”

Remark #5:

“P. 3: Spell out MSigDB”

Reply: We corrected the spelling in our manuscript.

Remark #6:

“Mention examples on what it means that: a gene is likely to play a role in that gene set”

Reply: We changed the following sentences in our Results section on Page 4, Line 19 :

“Strong positive correlation indicates that the transcriptional regulation of an individual gene is similar to the average transcriptional regulation of members of the gene set. Gene sets that capture biological processes with highly co-regulated genes generate consistent regulatory barcodes. Therefore, genes with strong correlation have a high likelihood of belonging in those gene sets even if they are not currently members. For example, *BRCA2* is a DNA repair protein with a strong co-regulation with the cell cycle KEGG gene set (Prediction score = 15.53). Even though *BRCA2* does not directly regulate the cell cycle, progression through the G2–M checkpoint depends on the successful repair of all DNA damaged during replication. Therefore, when viewed through co-regulation *BRCA2* participates in two biological processes namely, DNA repair and cell cycle progression. ”

Remark #7:

“Table not properly described. For instance, it is not clearly stated that GENETICA is the method proposed by the authors”

Reply: We changed the following Table:

Table 1: Guilt by association methods

	DEPICT and GADO	GENETICA (ours)
Strategy	Guilt by association prediction of gene functions using gene sets	Guilt by association prediction of gene functions using gene sets
Transcriptional Components	Principal component analysis	Independent component analysis
Barcode	Comparison of eigenvector scalars of member genes and non-member genes using Welch T-test	Average of all mixing matrix weights of member genes
Distance metric	Pearson Correlation	Distance Correlation
Permutation test	No, parametric p-values used	Yes, permutation p-values used
Reference	[7, 8]	This study

References

- 1 Wang, S. *et al.* Genome-wide investigation of genes regulated by ER-alpha in breast cancer cells. *Molecules* **23** (2018).
- 2 Parnas, O. *et al.* A Genome-wide CRISPR screen in primary immune cells to dissect regulatory networks. *Cell* **162**, 675–86 (2015).
- 3 Lenk, G. M. *et al.* CRISPR knockout screen implicates three genes in lysosome function. *Sci Rep* **9**, 9609 (2019).
- 4 Flint, M. *et al.* A genome-wide CRISPR screen identifies N-acetylglucosamine-1-phosphate transferase as a potential antiviral target for Ebola virus. *Nat Commun* **10**, 285 (2019).
- 5 Jiang, Y. *et al.* An expanded evaluation of protein function prediction methods shows an improvement in accuracy. *Genome Biol* **17**, 184 (2016).
- 6 Kovács, I. A. *et al.* Network-based prediction of protein interactions. *Nat Commun* **10**, 1240 (2019).
- 7 Pers, T. H. *et al.* Biological interpretation of genome-wide association studies using predicted gene functions. *Nat Commun* **6**, 5890 (2015).

- 8 Deelen, P. *et al.* Improving the diagnostic yield of exome-sequencing by predicting gene-phenotype associations using large-scale gene expression analysis. *Nat Commun* **10**, 1–13 (2019).
- 9 Azodi, C. B., Pardo, J., VanBuren, R., de Los Campos, G. & Shiu, S. H. Transcriptome-based prediction of complex traits in maize. *Plant Cell* **32**, 139–51 (2020).
- 10 Keilwagen, J., Hartung, F., Paulini, M., Twardziok, S. O. & Grau, J. Combining RNA-seq data and homology-based gene prediction for plants, animals and fungi. *BMC Bioinformatics* **19**, 189 (2018).
- 11 Zhao, C. & Wang, Z. GOGO: An improved algorithm to measure the semantic similarity between gene ontology terms. *Sci Rep* **8**, 15107 (2018).
- 12 Kinalis, S., Nielsen, F. C., Winther, O. & Bagger, F. O. Deconvolution of autoencoders to learn biological regulatory modules from single cell mRNA sequencing data. *BMC Bioinformatics* **20**, 379 (2019).
- 13 Grønbech, C. H. *et al.* scVAE: variational auto-encoders for single-cell gene expression data. *Bioinformatics* **36**, 4415–22 (2020).
- 14 Sureyya Rifaioğlu, A., Dogan, T., Jesus Martin, M., Cetin-Atalay, R. & Atalay, V. DEEPred: Automated protein function prediction with multi-task feed-forward deep neural networks. *Sci Rep* **9**, 7344 (2019).
- 15 Cai, Y., Wang, J. & Deng, L. SDN2GO: An integrated deep learning model for protein function prediction. *Front Bioeng Biotechnol* **8**, 391 (2020).
- 16 Dorrity, M. W., Saunders, L. M., Queitsch, C., Fields, S. & Trapnell, C. Dimensionality reduction by UMAP to visualize physical and genetic interactions. *Nat Commun* **11**, 1537 (2020).
- 17 Nikolsky, Y. *et al.* Genome-wide functional synergy between amplified and mutated genes in human breast cancer. *Cancer Res* **68**, 9532–40 (2008).

- 18 Yeoh, A. E. *et al.* Effective Response Metric: a novel tool to predict relapse in childhood acute lymphoblastic leukaemia using time-series gene expression profiling. *Br J Haematol* **181**, 653–63 (2018).
- 19 Takahashi, E. *et al.* Tumor necrosis factor-alpha regulates transforming growth factor-beta-dependent epithelial-mesenchymal transition by promoting hyaluronan-CD44-moesin interaction. *J Biol Chem* **285**, 4060–73 (2010).
- 20 McInnes, L., Healy, J., Saul, N. & Grossberger, L. Umap: Uniform manifold approximation and projection. *J. Open Source Softw* **3**, 861 (2018).
- 21 Bray, N. L., Pimentel, H., Melsted, P. & Pachter, L. Near-optimal probabilistic RNA-seq quantification. *Nat Biotechnol* **34**, 525–27 (2016).
- 22 Love, M. I., Huber, W. & Anders, S. Moderated estimation of fold change and dispersion for RNA-seq data with DESeq2. *Genome Biol* **15**, 550 (2014).
- 23 Franzén, O., Gan, L. M. & Björkegren, J. L. M. PanglaoDB: a web server for exploration of mouse and human single-cell RNA sequencing data. *Database* **2019** (2019).
- 24 Li, Q., Birkbak, N. J., Györffy, B., Szallasi, Z. & Eklund, A. C. Jetset: selecting the optimal microarray probe set to represent a gene. *BMC Bioinformatics* **12**, 474 (2011).

Reviewers' Comments:

Reviewer #1:

Remarks to the Author:

The authors addressed all my concerns and I have no further questions. I recommend this manuscript for publication in Nat. Comm. journal.

Reviewer #3:

Remarks to the Author:

I would like to thank the authors' for addressing my comments, performing the additional analyses.

If I understand the authors correctly, then their intention with the added RNA-seq data analysis was to compare the microarray-based ICA-TCs to the RNA-seq-based ICA-TCs. It would be useful if the authors also could compute the ICA-TC based on the RNA-seq data without matching/limiting this analysis to the microarray analysis. In the RNA-seq data there are many more genes and the TCs from that analysis will be of interest to many other groups and hence the constructions of these TCs and the co-functionality predictions should not be limited by the microarray data.

Along the same lines and as previously noted, I think the ICA-based TCs will be of major use for several other groups. Therefore it would be great if the authors could make them available in a more FAIR (Findable, Accessible, Interoperable, Reusable) way. For instance, providing access to human and species-specific bulk download of all ICA-TCs (microarray-optimized and RNA-seq optimized; separate for each gene set databased on one combined across all databases), precise descriptions on the website how the given TC-dataset was constructed (in terms of samples, methods, version of gene sets etc) and provide the scripts used to generate the given set of TCs.

Finally, from a technical perspective, would it be possible to extract ICA-TCs simultaneously based on the microarray and RNA-seq data? Again, the better the construction of, access to and documentation of the ICA-TCs the more useful the work would be to the field.

Please note that the link to the figures etc. does not work (<https://doi.org/10.6084/m9.%0Cgshare.13265159>).

To summarize, the work by Urzua-Traslavina et al. provides useful resources to the community, however, it would be of even greater value if the authors could provide access to RNA-seq-based and microarray-based TCS in a more FAIR way.

Apologies for my delay in getting back to you.

All the best

Tune H. Pers

Urzua-Traslavina *et al.*: Improving gene function predictions using independent transcriptional components (NCOMMS–20–18207A)

We would like to thank both reviewers for their time invested in our response. Below, please find a point-by-point reply.

Reviewer 1

Remark #1:

“The authors addressed all my concerns and I have no further questions. I recommend this manuscript for publication in Nat. Comm. journal.”

Reply: We are glad that all the concerns of reviewer 1 were addressed. We are grateful for the recommendation for publication in Nature Communications.

Reviewer 3

We would like to apologize for the lack of accessibility of our datasets in the website that were stated to be there in the manuscript. We have corrected this oversight and provide more details in this response.

Remark #1:

“If I understand the authors correctly, then the intention with the added RNA-seq data analysis was to compare the microarray-based ICA-TCs to the RNA-seq-based ICA-TCs. It would be useful if the authors also could compute the ICA-TC based on the RNA-seq data without matching/limiting this analysis to the microarray analysis. ”

Reply: We completely agree some genes such as non-coding genes are only available for the RNA-seq profiles. Obtaining predictions for these non-coding genes was also the main goal of our analysis. We would like to reassure the reviewer that ICA-TC based gene predictions obtained with the RNA-seq profiles were only subsetted to the microarray matching subset for comparison purposes. By contrast, the website showcases all predictions generated for each input dataset. For example *RNVU1-8* is a non-coding gene and only has predictions available when using RNA-seq profiles. We would like to note that the gene perspective and gene-set perspective tabs in the website can be used to search the predictions of non-coding genes only in RNA-seq version (Clicking RNA-seq version in the header of the website).

We amended the following sentences in our Supplementary Notes on Page 9 line 20:

“To compare predictions obtained between RNA-seq and microarray datasets, Ensembl gene IDs were mapped to entrez IDs.”

Remark #2:

“I think the ICA-based TCs will be of major use for several other groups. Therefore it would be great if the authors could make them available in a more FAIR (Findable, Accessible, Interoperable, Reusable) way. For instance, providing access to human and species-specific bulk download of all ICA-TCs (microarray-optimized and RNA-seq optimized; separate for each gene set databased on one combined across all databases), precise descriptions on the website how the given TC-dataset was constructed (in terms of samples, methods, version of gene sets etc) and provide the scripts used to generate the given set of TCs.”

Reply: Again, we apologize for an oversight that resulted in the download links not being available in the website. We provide for the human microarray, mouse microarray and RNA-seq profiles the following: metadata, independent components, mixing matrix, gene set collections used to generate predictions, transcriptional barcodes for every gene set and prediction score tables for every gene set collection. This is findable in the Support Data tab in the website corresponding to the active dataset version selected in the website header. Additionally, we provide in the About tab of the website a link to the executable program used to run consensus independent component analysis and to generate prediction scores. We provide a README that explains how to set up and reproduce our results.

We would like to warn that modern browsers may sometimes temporarily save a cache of the nonupdated website which can prevent the display of the updated website with the Support Data tab. To bypass this issue without waiting for a cache update, please clear the cache of the browser or open the website on a different browser or computer that was not previously used to open the nonupdated website.

Remark #3:

“Finally, from a technical perspective, would it be possible to extract ICA-TCs simultaneously based on the microarray and RNA-seq data?”

Reply: It is technically possible to append samples from different platforms and perform independent component analysis on such hybrid dataset as long as the set of genes is reduced to the matching subset between datasets. However, because the human microarray ($n = 106,462$) contains more samples than the RNA-seq dataset ($n = 29,138$) this could bias the gene-gene covariance matrix towards the microarray profiles and therefore bias the determination of transcriptional components. Although this bias could potentially be balanced by downsampling the microarray dataset, the reduced sample size would miss the goal of performing such joint analysis. Since the RNA-seq dataset provides comparable prediction scores for more genes using only a third of the samples we believe that running a

RNA-seq microarray hybrid dataset would only provide marginal improvements in the best case scenario.

Remark #4:

“Please note that the link to the figures etc. does not work ”

Reply: We apologize if this prevented access to our figure data. We assure the reviewer that the link is not broken or unresponsive concerning the Figshare service. We suspect that the link address may become corrupted by the MTS since we cannot reproduce the error ourselves. We are now sharing the Figshare link in the cover letter and in this response to have multiple options in case one becomes broken. If all of the above fails, please search for the co-author “Vincent Leeuwenburgh” in the Figshare search field. Our dataset should appear first and is titled: “Improving gene function predictions using independent transcriptional components - Raw Figure Data”. Here is the link for convenience: <https://doi.org/10.6084/m9.figshare.13265159.v1>

Reviewers' Comments:

Reviewer #3:

Remarks to the Author:

All set. Thank you for the additional clarifications and making the data publicly available.

Best wishes

Tune H Pers.